**Data Availability Statement:** All relevant data are within the manuscript and its Supporting Information files.

# Suicide among physicians and health-care workers: A systematic review and meta-analysis

**Frédéric Dutheil**[1,2]☯*, **Claire Aubert**[3]☯, **Bruno Pereira**[4], **Michael Dambrun**[5], **Fares Moustafa**[6], **Martial Mermillod**[7,8], **Julien S. Baker**[9], **Marion Trousselard**[10], **François-Xavier Lesage**[11], **Valentin Navel**[12]

1 Université Clermont Auvergne, CNRS, LaPSCo, Physiological and Psychosocial Stress, CHU Clermont-Ferrand, University Hospital of Clermont-Ferrand, Occupational and Preventive Medicine, WittyFit, Clermont-Ferrand, France, 2 Australian Catholic University, Faculty of Health, School of Exercise Science, Melbourne, Victoria, Australia, 3 Université de Versailles Saint-Quentin-en-Yvelines, Faculty of Health Science Simone Veil, Versailles, France, 4 CHU Clermont-Ferrand, University Hospital of Clermont-Ferrand, Biostatistics Unit, the Clinical Research and Innovation Direction, Clermont-Ferrand, France, 5 Université Clermont Auvergne, CNRS, LaPSCo, Physiological and Psychosocial Stress, Clermont-Ferrand, France, 6 CHU Clermont-Ferrand, University Hospital of Clermont-Ferrand, Emergency, Clermont-Ferrand, France, 7 Univ. Grenoble Alpes, Univ. Savoie Mont Blanc, CNRS, LPNC, Grenoble, France, 8 Institut Universitaire de France, Paris, France, 9 Centre for Health and Exercise Science Research, Department of Sport, Physical Education and Health, Hong Kong Baptist University, Kowloon Tong, Hong Kong, 10 French Armed Forces Biomedical Research Institute-IRBA, Neurophysiology of Stress, Neuroscience and Operational Constraint Department, Brétigny-sur-Orge, France, 11 University of Montpellier, Laboratory Epsylon EA, Dynamic of Human Abilities & Health Behaviors, CHU Montpellier, University Hospital of Montpellier, Occupational and Preventive Medicine, Montpellier, France, 12 CHU Clermont-Ferrand, University Hospital of Clermont-Ferrand, Ophthalmology, Clermont-Ferrand, France

☯ These authors contributed equally to this work.

* frederic.dutheil@uca.fr

## Abstract

### Background

Medical-related professions are at high suicide risk. However, data are contradictory and comparisons were not made between gender, occupation and specialties, epochs of times. Thus, we conducted a systematic review and meta-analysis on suicide risk among health-care workers.

### Method

The PubMed, Cochrane Library, Science Direct and Embase databases were searched without language restriction on April 2019, with the following keywords: suicide* AND (« health care worker* » OR physician* OR nurse*). When possible, we stratified results by gender, countries, time, and specialties. Estimates were pooled using random-effect meta-analysis. Differences by study-level characteristics were estimated using stratified meta-analysis and meta-regression. Suicides, suicidal attempts, and suicidal ideation were retrieved from national or local specific registers or case records. In addition, suicide attempts and suicidal ideation were also retrieved from questionnaires (paper or internet).

**Funding:** The authors received no specific funding for this work.

**Competing interests:** The authors have declared that no competing interests exist.

## Results

The overall SMR for suicide in physicians was 1.44 (95CI 1.16, 1.72) with an important heterogeneity ($I^2$ = 93.9%, p<0.001). Female were at higher risk (SMR = 1.9; 95CI 1.49, 2.58; and ES = 0.67; 95CI 0.19, 1.14; p<0.001 compared to male). US physicians were at higher risk (ES = 1.34; 95CI 1.28, 1.55; p <0.001 vs Rest of the world). Suicide decreased over time, especially in Europe (ES = -0.18; 95CI -0.37, -0.01; p = 0.044). Some specialties might be at higher risk such as anesthesiologists, psychiatrists, general practitioners and general surgeons. There were 1.0% (95CI 1.0, 2.0; p<0.001) of suicide attempts and 17% (95CI 12, 21; p<0.001) of suicidal ideation in physicians. Insufficient data precluded meta-analysis on other health-care workers.

## Conclusion

Physicians are an at-risk profession of suicide, with women particularly at risk. The rate of suicide in physicians decreased over time, especially in Europe. The high prevalence of physicians who committed suicide attempt as well as those with suicidal ideation should benefits for preventive strategies at the workplace. Finally, the lack of data on other health-care workers suggest to implement studies investigating those occupations.

## Introduction

Suicide risk was increased in certain occupational groups, especially in medical-related professions [1]. Physicians, and other health-care workers such as nurses [2,3], were considered like high risk group of suicide in different countries [4,5,6], especially for women [6,7,8]. Indeed, despite considerably higher risk of suicides in men than women in the general population [9], female doctors have higher suicide rates than men [10], putatively because of their social family role [11], or a poor status integration within the profession [7]. Suicide rate in physicians was also not homogenous in all countries [12], and physicians' satisfaction has been reported to change between different epochs of times [13]. Physicians working conditions varied substantially between countries and over contemporary times, these factors were never investigated in relationships with suicide in physicians. For example, there were tentative to regulate working time of physicians over the recent years, such as in Europe with its European Working Time Directive (EWTD) [14]. Some specialties have been suggested to be particularly at risk of suicides [15,16] with occupational factors individualized in different medical or surgical specialties: heavy workload and working hours involved in the job such as long shifts and unpredictable hours (with the sleep deprivation associated) [17], stress of the situations (life and death emergencies) [18], and easy access to a means of committing suicide [19]. To implement coordinated and synergistic preventive strategies, we need to identify physicians in mental health suffering [20], therefore statistical analyses on suicide attempts and suicidal ideation were necessary. However, robust statistics on health-care workers were desperately lacking for suicides, suicide attempts and suicidal ideation. The latest meta-analysis summarized physicians suicide risk before 2000s [6], we need for updated synthesis of literature. We hypothesized that 1) physicians are more at risk to commit suicide than the general population, 2) women physicians are more at risk to commit suicide than their male counterparts, 3) some countries would have higher rates of suicide in physicians, 4) with an improvement over time, 5) some medical or surgical specialties would be at higher risk of suicide, 6) physicians would also exhibit higher rates of suicide attempts and suicidal ideation, and 7) other health care workers would also be at risk of suicide.

Thus, we aimed to conduct a systematic review of the literature and meta-analysis to provide evidence-based data for suicide risk among health-care workers, considering gender, geographic zone, epoch of time, medical and surgical specialties. Finally, we wanted to expand our study to suicide attempts and suicidal ideation.

## Methods

### Search strategy and study eligibility

We reviewed all studies involving suicides, suicide attempts or suicidal ideation in health-care workers. Students were excluded because of the difference in responsibilities in comparisons with health-care workers, and because of the existence of previous recent meta-analyses focusing specifically on health-care students [21,22,23,24]; we included interns because they were not included in the aforementioned meta-analyses on prevalence of suicides, suicide attempts or suicidal ideation, and because they could have similar responsibilities to senior practitioners. The PubMed, Cochrane Library, Science Direct and Embase databases were searched on April 2019, with the following keywords: suicide* AND (« health care worker* » OR physician* OR nurse*). The search was not limited by years or languages. To be included, articles had to be peer-reviewed and to describe original empirical data on suicides, suicide attempt or suicidal ideation in health-care workers. When data were available, we also collected data from a control group (such as general population) for comparisons purposes. In addition, reference lists of all publications meeting the inclusion criteria will be manually searched to identify any further studies not found through digital research. The search strategy was presented in Fig 1. Three authors (Claire Aubert, Valentin Navel and Frederic Dutheil) conducted all literature searches, and separately reviewed the abstracts and decided the suitability of the articles for inclusion. Two others authors (Bruno Pereira and Martial Mermillod) have been asked to review the articles when consensus on suitability was debated. Then all authors reviewed the eligible articles.

### Quality of assessment

Although not designed for quantifying the integrity of studies [25], the "STrengthening the Reporting of Observational studies in Epidemiology" (STROBE) criteria [26] and Newcastle-Ottawa Scale (NOS) were used to check the quality of articles [27]. The maximum score in STROBE criteria was 30 with assessment of 22 items, in NOS criteria was 9 with assessment of 8 items (one star for each item within the selection and exposure category and a maximum of two stars for comparability) (Figs 2 and 3).

## Statistical considerations

Statistical analysis was conducted using Comprehensive Meta-analysis software (version 2, Biostat Corporation) [28,29,30] and Stata software (version 13, StataCorp, College Station, US) [28,29,31]. Main characteristics were summarized for each study sample and reported as mean (standard-deviation) and number (%) for continuous and categorical variables respectively. Statistical heterogeneity between results was assessed by examining forest plots, confidence intervals (CI) and using formal tests for homogeneity based on $I^2$ statistic, which is the most common metric for measuring the magnitude of heterogeneity between studies and is easily interpretable. $I^2$ values range between 0% and 100% and are typically considered low for <25%, moderate for 25–50%, and high for > 50%. Random effect meta-analysis (DerSimonian and Liard approach) were conducted when data could be pooled [32]. P values < 0.05 were considered statistically significant. We conducted: 1) meta-analyses on the Standardized Mortality Ratio (SMR) for suicides i.e. the ratio between the observed and expected number of

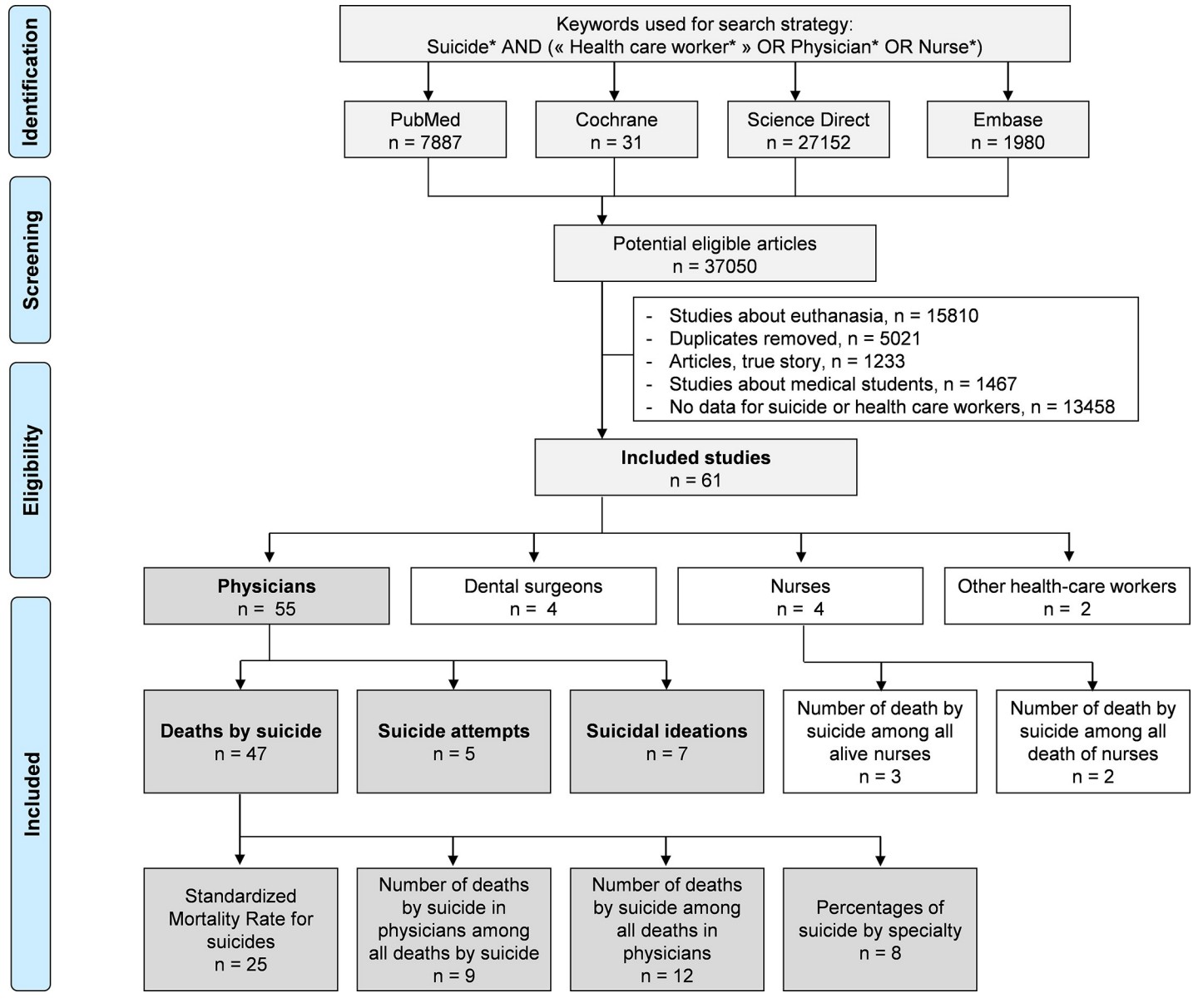

**Fig 1. Search strategy.**

death among physicians, stratified by sex (Fig 4; and Fig 5 for metaregressions), geographic zones (Fig 6), epochs of time, and by categories of specialties (main groups of specialities (Fig 7 and S1 Fig), surgical specialties (Fig 8 and S2 Fig), then medical specialities (Fig 9 and S3 Fig), 2) meta-analyses on the prevalence of health-care workers died by suicide among all health-care workers death (Fig 10), 3) meta-analyses on the prevalence of health-care workers died by suicide among all the deaths by suicide in the general population (S4 Fig), 4) meta-analyses on suicide attempts (S5 Fig) and suicidal ideation (Fig 11). Effect-size was estimated for quantitative endpoints as number of physicians having done suicide attempt and number of physicians with suicidal ideation. A scale for ES has been suggested with 0.8 reflecting a large effect, 0.5 a moderate effect, and 0.2 a small effect [33]. When possible (sufficient sample size), meta-regressions were proposed to study relation between prevalence and epidemiological relevant

Methodological quality of included articles using Newcastle – Ottawa Quality Assessment Scale
Yes:+
No: -
Can't say: ?
Not applicable: NA

| | Selection bias | | | | Comparability bias | | Outcome bias | | |
|---|---|---|---|---|---|---|---|---|---|
| | Representativeness of the exposed | Selection of the non exposed | Ascertainment of exposed | Outcome of interest was no present at start | Study controls for the most important factor | Study controls for any important factor | Assessment of outcome | follow-up long enough | Adequacy of follow up |
| Aasland 2001 | + | + | + | + | + | + | + | + | + |
| Aasland 2011 | + | + | + | + | + | + | + | + | + |
| Arnetz 1987 | + | + | + | + | + | ? | + | + | + |
| Austin 2013 | + | - | + | + | + | NA | + | - | + |
| Baymar 1986 | + | + | + | + | + | ? | + | + | + |
| Brooks 2017 | - | + | - | + | + | + | - | + | + |
| Carpenter 1997 | + | + | + | + | + | - | + | + | + |
| Craig 1968 | + | + | + | + | + | + | + | + | + |
| Davidson 2018 | + | + | + | + | + | + | + | + | + |
| Dean 1969 | + | + | + | + | + | + | - | + | + |
| Desole 1969 | + | NA | + | + | NA | NA | + | + | + |
| Everson 1975 | + | NA | + | + | + | + | - | + | + |
| Franck 1999 | - | + | - | - | + | + | + | NA | - |
| Franck 2000 | + | + | + | + | + | + | + | + | + |
| Fridner 2009 | + | + | - | NA | + | ? | - | + | - |
| Gagne 2011 | + | + | + | + | + | + | + | + | + |
| Gold 2013 | + | + | + | + | + | + | + | + | + |
| Gunnarsdottir 1995 | + | + | + | + | + | ? | + | + | + |
| Hawton 2001 | + | + | + | + | + | + | + | + | + |
| Hawton 2002 | + | + | + | + | + | + | + | + | - |
| Hawton 2011 | + | + | + | + | + | + | + | + | + |
| Hem 2000 | - | + | - | + | + | ? | - | + | - |
| Hem 2005 | + | + | + | + | + | + | + | + | + |
| Hemenway 1993 | - | ? | + | + | + | ? | - | + | - |
| Herner 1993 | NA | NA | NA | NA | NA | NA | NA | NA | NA |
| Hikiji 2013 | + | + | + | + | + | + | + | + | + |
| Hubbard 1922 | NA | NA | NA | NA | NA | NA | NA | NA | NA |
| Innos 2002 | - | + | + | + | + | + | + | + | + |
| Jones 1977 | - | NA | + | + | + | + | - | + | + |
| Juel 1999 | + | + | + | + | + | + | + | + | + |
| Lew 1979 | + | NA | + | + | + | + | + | + | + |
| Linde 1981 | + | + | + | + | + | + | + | + | + |
| Lindeman 1997 | + | + | + | + | + | + | + | + | + |
| Lindeman 2007 | + | + | + | + | + | ? | + | - | + |
| Lindfors 2009 | + | ? | + | + | + | + | - | + | - |
| Lindhardt 1963 | + | + | + | + | + | ? | + | + | + |
| Loas 2018 | + | + | - | + | + | NA | - | + | - |
| Mintz 2018 | + | + | + | + | + | + | + | + | + |
| No author 1986 | + | + | + | + | + | + | + | - | - |
| Nordentoft 1988 | NA | NA | NA | NA | NA | NA | NA | NA | NA |
| Olkinuora 1990 | + | + | + | + | + | + | + | + | + |
| Palhares-Alves 2015 | + | + | + | + | + | + | + | + | + |
| Petersen 2008 | + | + | + | + | + | + | + | + | + |
| Pitts 1979 | + | + | + | + | + | + | + | + | + |
| Rafnsson 1998 | NA | NA | NA | NA | NA | NA | NA | NA | NA |
| Revicki 1985 | + | + | + | + | + | + | + | + | + |
| Rich 1979 | + | + | + | + | + | + | + | + | + |
| Rich1980 | + | ? | + | + | + | + | + | + | + |
| Rimpela 1987 | + | + | + | + | + | + | + | + | + |
| Rose 1973 | + | ? | + | + | + | + | + | + | + |
| Roy 1985 | NA | NA | NA | NA | NA | NA | NA | NA | NA |
| Samkoff 1995 | + | + | + | + | + | - | + | + | + |
| Schlicht 1990 | + | + | + | + | + | + | + | + | + |
| Shang 2011 | + | + | + | + | + | + | + | + | + |
| Shang 2012 | + | + | + | ? | + | ? | + | + | + |
| Simon 1968 | - | - | + | + | + | - | - | + | + |
| Stefansson 1991 | + | + | + | + | + | + | + | + | + |
| Torre 2005 | + | + | + | + | + | + | + | + | + |
| Ulmann 1991 | + | + | + | + | + | + | + | + | + |
| Wang 2017 | - | - | - | + | ? | ? | - | + | ? |
| Zang 2018 | + | + | + | + | + | + | + | + | + |

**Fig 2. Methodological quality of included articles using Newcastle–Ottawa Quality Assessment Scale.**

parameters determined according to the literature: sex, geographic zone, epoch of time (for studies with a follow-up over several consecutive years, we based our statistics on the mean year of epoch of time). Results were expressed as regression coefficient and 95% CI.

## Results

An initial search produced a possible 37050 articles (Fig 1). Removal of duplicates and use of the selection criteria reduced the search to 61 articles [1,2,5,7,8,15,16,34,35,36,37,38,39,40, 41,42,43,44,45,46,47,48,49,50,51,52,53,54,55,56,57,58,59,60,61,62,63,64,65,66,67,68,69,70, 71,72,73,74,75,76,77,78,79,80,81,82,83,84,85,86,87]. In those 61 articles, 55 articles were on physicians [1,5,7,8,15,16,34,35,36,37,38,39,40,41,42,43,44,45,46,47,48,49,50,51,52,53,54,55, 56,57,58,59,60,61,62,63,64,65,66,67,68,69,70,71,72,73,74,75,76,77,78,82,83,84,85], four on dental surgeons [55,56,62,70], four on nurses [2,79,80,86], and two on other health-care workers [70,87]. Among those 55 on physicians, 47 reported data on deaths by suicide [1,5,7, 8,15,16,34,35,36,37,38,39,40,41,42,43,44,45,46,47,48,49,50,51,52,53,54,55,56,57,58,59,60, 61,62,63,64,65,66,67,68,69,70,71,72,82,83], five on suicide attempts [47,73,75,77,85], and seven on suicidal ideation [74,75,76,77,78,84,85]. In those 47 articles on deaths by suicide among physicians, 25 described SMR for suicide [7,8,41,46,52,54,55,56,57,58,59,60,61,62,63, 64,65,66,67,68,69,70,71,72,82], eight reported percentages of suicide by specialty [15,16,40, 43,45,47,51,83], 12 reported the number of physicians died by suicide among all deaths in

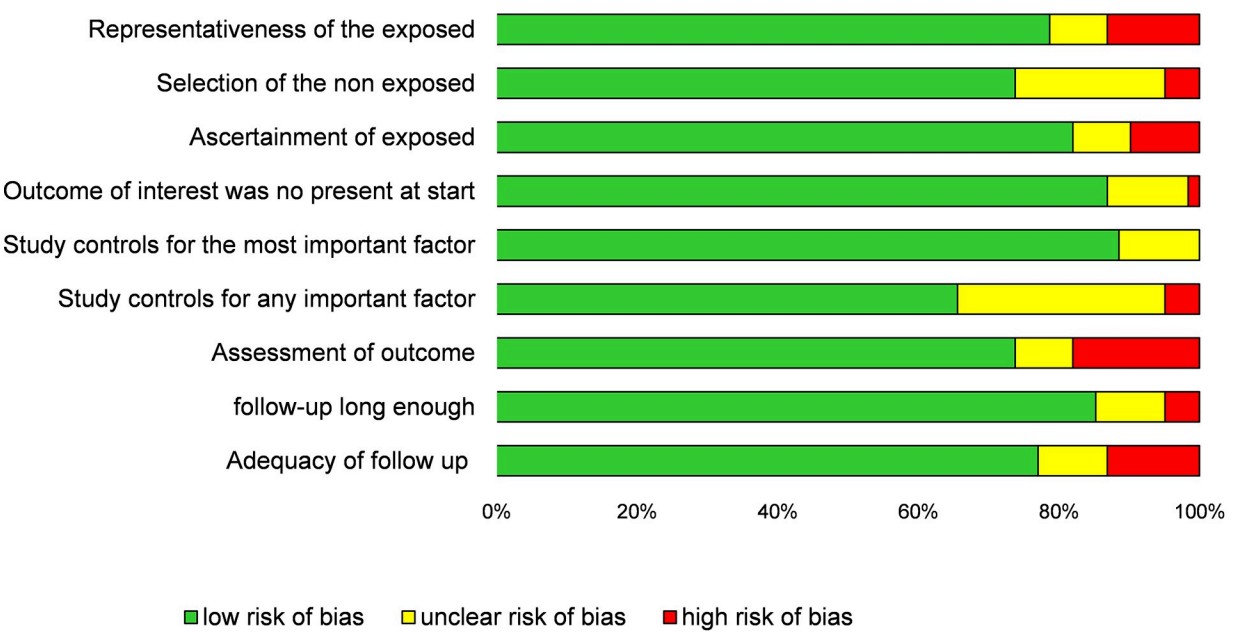

**Fig 3. Summary bias risk of included articles using the Newcastle–Ottawa Quality Assessment Scale model.**

| Study | Physicians n of death by suicides | Sex* (%male) | Country | Time (period of observation) | | SMR (95%CI) | | Weight (%) |
|---|---|---|---|---|---|---|---|---|
| **Men** | | | | | | | | |
| Arnetz 1987 | 42 | 76% | Europe | 1961-1970 | | 1.20 | (0.85, 1.69) | 3.4 |
| Baymar 1986 | 94 | 72% | Europe | 1963-1978 | | 1.58 | (1.07, 2.34) | 3.1 |
| Carpenter 1997 | 64 | 87% | Europe | 1962-1979 | | 0.96 | (0.72, 1.25) | 3.5 |
| Dean 1969 | 23 | 96% | Africa | 1960-1966 | | 1.26 | (0.74, 2.13) | 3.0 |
| Frank 2000 | 416 | 91% | North America | 1984-1995 | | 1.70 | (1.53, 1.88) | 3.6 |
| Hawton 2001 | 57 | 74% | Europe | 1991-1995 | | 0.67 | (0.47, 0.87) | 3.6 |
| Herner 1993 | 25 | 68% | Europe | 1989-1991 | | 1.10 | (0.80, 1.52) | 3.4 |
| Innos 2002 | 11 | 54% | Europe | 1983-1998 | | 0.58 | (0.21, 1.27) | 3.2 |
| Juel 1999 | 194 | 86% | Europe | 1973-1992 | | 1.64 | (1.40, 1.91) | 3.5 |
| Lindeman 1997 | 51 | 70% | Europe | 1986-1993 | | 0.87 | (0.69, 1.10) | 3.6 |
| Lindhardt 1963 | 67 | 100% | Europe | 1935-1959 | | 1.53 | (1.06, 2.20) | 3.2 |
| Nordentoft 1988 | 69 | 85% | Europe | 1970-1980 | | 2.46 | (1.31, 4.60) | 1.6 |
| Petersen 2008 | 203 | 89% | North America | 1984-1992 | | 0.80 | (0.53, 1.20) | 3.5 |
| Rafnsson 1998 | 7 | 100% | Europe | 1955-1995 | | 1.01 | (0.40, 2.04) | 2.8 |
| Revicki 1985 | 13 | 100% | North America | 1978-1982 | | 1.16 | (0.80, 1.70) | 3.3 |
| Rich 1979 | 544 | 100% | North America | 1967-1972 | | 1.03 | (0.74, 1.45) | 3.4 |
| Rimpela 1987 | 17 | 100% | Europe | 1971-1980 | | 1.28 | (1.00, 1.65) | 3.5 |
| Rose 1973 | 49 | 98% | North America | 1959-1961 | | 2.03 | (1.29, 3.19) | 2.5 |
| Schlicht 1990 | 13 | 77% | Australia | 1950-1986 | | 1.13 | (0.54, 2.07) | 2.8 |
| Stefansson 1991 | 138 | 82% | Europe | 1971-1985 | | 1.82 | (1.19, 2.80) | 2.8 |
| Torre 2005 | 22 | 91% | North America | 1948-1998 | | 1.82 | (1.11, 2.82) | 2.7 |
| Ullmann 1991, Loma Linda Univ | 46 | 100% | North America | 1910-1981 | | 1.48 | (0.97, 2.27) | 3.0 |
| Ullmann 1991, Univ of Southern California | 39 | 100% | North America | 1910-1981 | | 2.18 | (1.10, 4.32) | 1.6 |
| **Sub-total** (I2=79.1%, p<0.001) | | | | | | **1.24** | **(1.05, 1.43)** | **70.6** |
| **Women** | | | | | | | | |
| Arnetz 1987 | 42 | 76% | Europe | 1961-1970 | | 5.70 | (1.68, 10.72) | 0.3 |
| Baymar 1986 | 94 | 72% | Europe | 1963-1978 | | 2.96 | (1.44, 6.09) | 1.0 |
| Carpenter 1997 | 64 | 87% | Europe | 1962-1979 | | 2.15 | (0.93, 4.23) | 1.6 |
| Frank 2000 | 416 | 91% | North America | 1984-1995 | | 2.38 | (1.68, 3.28) | 2.8 |
| Hawton 2001 | 57 | 74% | Europe | 1991-1995 | | 2.02 | (1.00, 3.04) | 2.4 |
| Herner 1993 | 25 | 68% | Europe | 1989-1991 | | 2.32 | (1.12, 4.81) | 1.4 |
| Innos 2002 | 11 | 54% | Europe | 1983-1998 | | 0.62 | (0.20, 1.45) | 3.1 |
| Juel 1999 | 194 | 86% | Europe | 1973-1992 | | 1.68 | (1.10, 2.46) | 3.0 |
| Lindeman 1997 | 51 | 70% | Europe | 1986-1993 | | 2.33 | (1.08, 5.05) | 1.3 |
| Nordentoft 1988 | 69 | 85% | Europe | 1970-1980 | | 3.33 | (0.42, 26.29) | 0.1 |
| Petersen 2008 | 203 | 89% | North America | 1984-1992 | | 2.39 | (1.52, 3.77) | 2.3 |
| Pitts 1979 | 49 | 0% | North America | 1967-1972 | | 3.57 | (1.23, 10.40) | 0.3 |
| Schlicht 1990 | 13 | 77% | Australia | 1950-1986 | | 5.01 | (1.01, 14.65) | 0.2 |
| Stefansson 1991 | 138 | 82% | Europe | 1971-1985 | | 5.02 | (1.67, 15.03) | 0.2 |
| Torre 2005 | 22 | 91% | North America | 1948-1998 | | 4.95 | (0.56, 17.85) | 0.1 |
| **Sub-total** (I2= 42.5%, p< 0.041) | | | | | | **1.94** | **(1.49, 2.58)** | **20.1** |
| **Men + Women** | | | | | | | | |
| Davidson 2018 | 38 | - | North America | 2005-2015 | | 2.29 | (1.66, 3.08) | 2.9 |
| Shang 2011 | 23 | - | Asia | 1990-2006 | | 0.14 | (0.09, 0.21) | 3.6 |
| **Sub-total** (I2= 97.1%, p< 0.001.) | | | | | | **1.19** | **(-0.92, 3.29)** | **6.5** |
| **Overall** (I2= 93.9%, p< 0.001) | | | | | | **1.44** | **(1.16, 1.72)** | **100** |

0    3,5    7

**Fig 4. Meta-analysis of standardized mortality rate for suicides among physicians by gender.**

physicians [16,39,41,42,44,46,48,49,50,51,52,53], and nine reported the number of physicians died by suicide among all the deaths by suicide in the general population [1,5,15,34,35,36,37, 38,82]. As there are few exploitable studies about dental surgeons, nurses and other health-care workers, we won't treat them in that meta-analysis.

More details on study characteristics (Table 1), quality of articles (Figs 2 and 3), method of sampling for markers analysis, inclusion and exclusion criteria, characteristics of participants, outcomes and aims of the studies, and study designs of included articles are described in S1 Appendix.

## Meta-analysis of the standardized mortality rate for suicides among physicians

We included 25 studies. The overall SMR was 1.44 (95CI 1.16, 1.72) with an important heterogeneity ($I^2$ = 93.9%). Among the 25 included studies, 17 studies reported both male and female

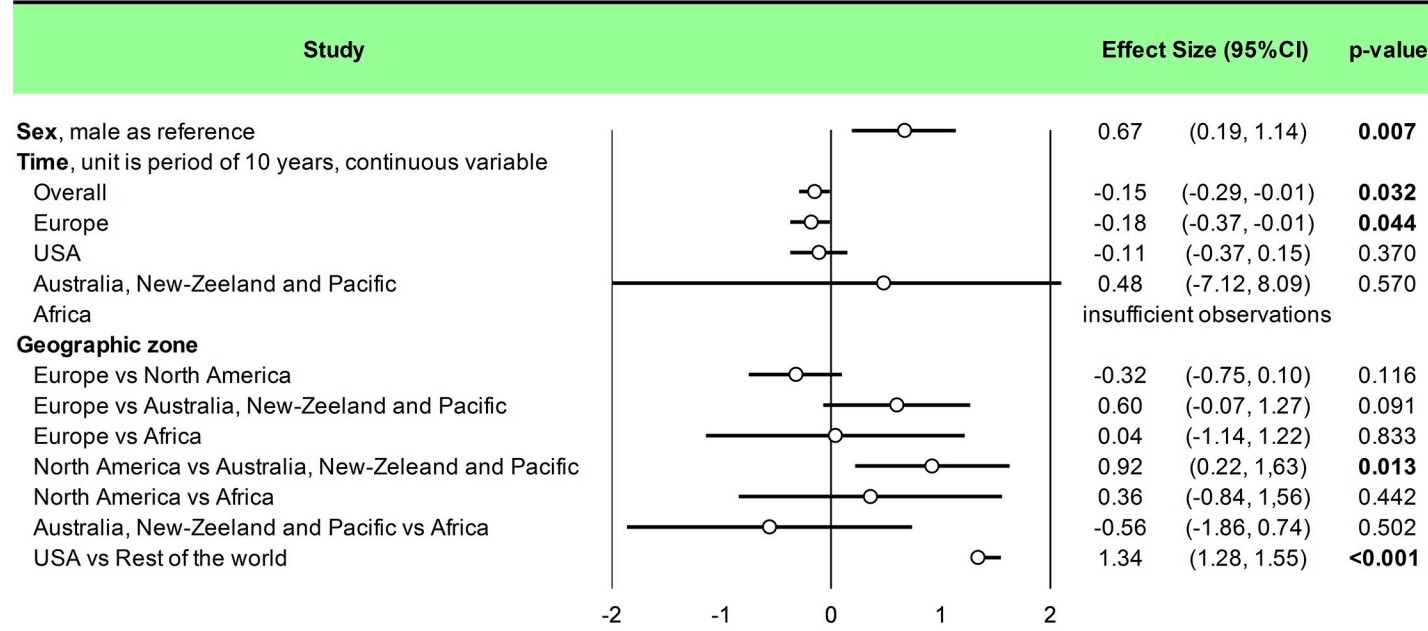

**Fig 5. Meta-regression of standardized mortality rate for suicides among physicians.**

physicians [7,8,41,46,52,54,55,56,57,58,59,61,62,68,70,71,82], six reported only male physicians [60,64,65,66,67,72], and one only reported female physicians [63]. We found a significantly higher risk of suicide among male physicians than in the general population (SMR = 1.24; 95CI 1.05, 1.43; $P < 0.001$; $I^2 = 79.1\%$) and for suicide among female physicians than in the general population (SMR = 1.94; 95CI 1.49, 2.58; $P < 0.041$; $I^2 = 42.5\%$) (Fig 4). Meta-regressions demonstrated that women physicians had a higher risk than their counterpart men to commit suicide (0.67; 95CI 0.19, 1.14; $P = 0.007$) (Fig 5). We further demonstrated that the risk of suicide was not homogeneous over all the countries. SMR was 1.27 (95CI 1.05, 1.49; $P < 0.001$; $I^2 = 71.3\%$) in Europe, 1.63 (95CI 1.29, 1.96; $P < 0.001$; $I^2 = 74.1\%$) in North America, 0.79 (95CI 0.03, 1.62; $P = 0.002$; $I^2 = 79.5\%$) in Australia, New-Zeeland and Pacific and 1.26 (95CI 0.56, 1.96) in Africa (Fig 6). Meta-regressions demonstrated a higher risk of suicide in North America than in Australia, New-Zeeland and Pacific (0.92; 95CI 0.22, 1.63; $P = 0.013$) and especially higher in USA vs the rest of the world (1.34; 95CI 1.28, 1.55; $P < 0.001$) (Fig 5).

Finally, we demonstrated an overall time effect (-0.15; 95CI -0.29, -0.01; $P = 0.032$) which signify that the risk decreased over time. This relationship is significant in Europe (-0.18; 95CI -0.37, -0.01; $P = 0.044$) but not in USA (-0.11; 95CI -0.37, 0.15; $P = 0.370$) or in Australia, New-Zeeland and Pacific (-0.48; 95CI -8.09, 7.12; $P = 0.570$). For Africa, there were insufficient observations (Fig 5).

## Meta-analysis of percentage of suicide in physicians by group of specialties

We included eight studies [15,16,40,43,45,47,51,83]. The percentage of suicide in general practitioners was 32% (95CI 21, 43; $P < 0.001$; $I^2 = 93.1\%$), in internal medicine was 16% (95CI 9, 23; $P < 0.001$; $I^2 = 88.6\%$), in psychiatrists was 11% (95CI 9, 14; $P = 0.30$; $I^2 = 17.5\%$), in other medical specialties was 3% (95CI 3, 4; $P = 0.02$; $I^2 = 40.7\%$), in surgeons was 4% (95CI 2, 5; $P < 0.001$; $I^2 = 62.8\%$) and in internships was 2% (95CI 1, 4) (Fig 7).

Meta-regressions demonstrated a higher risk of suicide in general practitioners than internal medicine (0.12; 95CI 0.05, 0.19; $P = 0.001$), than psychiatrists (0.17; 95CI 0.09, 0.24;

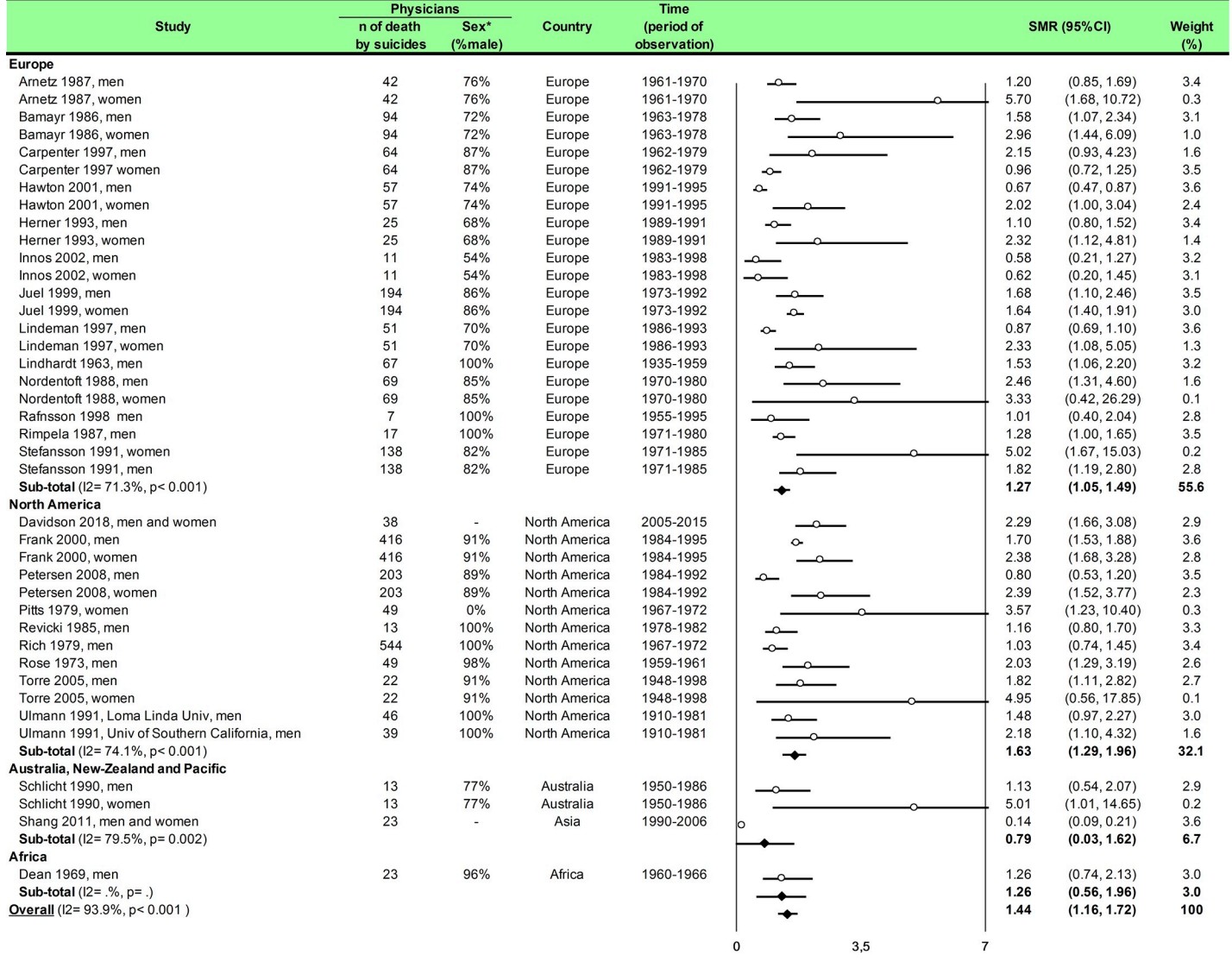

| | Physicians | | | Time | | SMR (95%CI) | | Weight |
|---|---|---|---|---|---|---|---|---|
| Study | n of death by suicides | Sex* (%male) | Country | (period of observation) | | | | (%) |
| **Europe** | | | | | | | | |
| Arnetz 1987, men | 42 | 76% | Europe | 1961-1970 | | 1.20 | (0.85, 1.69) | 3.4 |
| Arnetz 1987, women | 42 | 76% | Europe | 1961-1970 | | 5.70 | (1.68, 10.72) | 0.3 |
| Bamayr 1986, men | 94 | 72% | Europe | 1963-1978 | | 1.58 | (1.07, 2.34) | 3.1 |
| Bamayr 1986, women | 94 | 72% | Europe | 1963-1978 | | 2.96 | (1.44, 6.09) | 1.0 |
| Carpenter 1997, men | 64 | 87% | Europe | 1962-1979 | | 2.15 | (0.93, 4.23) | 1.6 |
| Carpenter 1997 women | 64 | 87% | Europe | 1962-1979 | | 0.96 | (0.72, 1.25) | 3.5 |
| Hawton 2001, men | 57 | 74% | Europe | 1991-1995 | | 0.67 | (0.47, 0.87) | 3.6 |
| Hawton 2001, women | 57 | 74% | Europe | 1991-1995 | | 2.02 | (1.00, 3.04) | 2.4 |
| Herner 1993, men | 25 | 68% | Europe | 1989-1991 | | 1.10 | (0.80, 1.52) | 3.4 |
| Herner 1993, women | 25 | 68% | Europe | 1989-1991 | | 2.32 | (1.12, 4.81) | 1.4 |
| Innos 2002, men | 11 | 54% | Europe | 1983-1998 | | 0.58 | (0.21, 1.27) | 3.2 |
| Innos 2002, women | 11 | 54% | Europe | 1983-1998 | | 0.62 | (0.20, 1.45) | 3.1 |
| Juel 1999, men | 194 | 86% | Europe | 1973-1992 | | 1.68 | (1.10, 2.46) | 3.5 |
| Juel 1999, women | 194 | 86% | Europe | 1973-1992 | | 1.64 | (1.40, 1.91) | 3.0 |
| Lindeman 1997, men | 51 | 70% | Europe | 1986-1993 | | 0.87 | (0.69, 1.10) | 3.6 |
| Lindeman 1997, women | 51 | 70% | Europe | 1986-1993 | | 2.33 | (1.08, 5.05) | 1.3 |
| Lindhardt 1963, men | 67 | 100% | Europe | 1935-1959 | | 1.53 | (1.06, 2.20) | 3.2 |
| Nordentoft 1988, men | 69 | 85% | Europe | 1970-1980 | | 2.46 | (1.31, 4.60) | 1.6 |
| Nordentoft 1988, women | 69 | 85% | Europe | 1970-1980 | | 3.33 | (0.42, 26.29) | 0.1 |
| Rafnsson 1998 men | 7 | 100% | Europe | 1955-1995 | | 1.01 | (0.40, 2.04) | 2.8 |
| Rimpela 1987, men | 17 | 100% | Europe | 1971-1980 | | 1.28 | (1.00, 1.65) | 3.5 |
| Stefansson 1991, women | 138 | 82% | Europe | 1971-1985 | | 5.02 | (1.67, 15.03) | 0.2 |
| Stefansson 1991, men | 138 | 82% | Europe | 1971-1985 | | 1.82 | (1.19, 2.80) | 2.8 |
| **Sub-total** (I2= 71.3%, p< 0.001) | | | | | | **1.27** | **(1.05, 1.49)** | **55.6** |
| **North America** | | | | | | | | |
| Davidson 2018, men and women | 38 | - | North America | 2005-2015 | | 2.29 | (1.66, 3.08) | 2.9 |
| Frank 2000, men | 416 | 91% | North America | 1984-1995 | | 1.70 | (1.53, 1.88) | 3.6 |
| Frank 2000, women | 416 | 91% | North America | 1984-1995 | | 2.38 | (1.68, 3.28) | 2.8 |
| Petersen 2008, men | 203 | 89% | North America | 1984-1992 | | 0.80 | (0.53, 1.20) | 3.5 |
| Petersen 2008, women | 203 | 89% | North America | 1984-1992 | | 2.39 | (1.52, 3.77) | 2.3 |
| Pitts 1979, women | 49 | 0% | North America | 1967-1972 | | 3.57 | (1.23, 10.40) | 0.3 |
| Revicki 1985, men | 13 | 100% | North America | 1978-1982 | | 1.16 | (0.80, 1.70) | 3.3 |
| Rich 1979, men | 544 | 100% | North America | 1967-1972 | | 1.03 | (0.74, 1.45) | 3.4 |
| Rose 1973, men | 49 | 98% | North America | 1959-1961 | | 2.03 | (1.29, 3.19) | 2.6 |
| Torre 2005, men | 22 | 91% | North America | 1948-1998 | | 1.82 | (1.11, 2.82) | 2.7 |
| Torre 2005, women | 22 | 91% | North America | 1948-1998 | | 4.95 | (0.56, 17.85) | 0.1 |
| Ulmann 1991, Loma Linda Univ, men | 46 | 100% | North America | 1910-1981 | | 1.48 | (0.97, 2.27) | 3.0 |
| Ulmann 1991, Univ of Southern California, men | 39 | 100% | North America | 1910-1981 | | 2.18 | (1.10, 4.32) | 1.6 |
| **Sub-total** (I2= 74.1%, p< 0.001) | | | | | | **1.63** | **(1.29, 1.96)** | **32.1** |
| **Australia, New-Zealand and Pacific** | | | | | | | | |
| Schlicht 1990, men | 13 | 77% | Australia | 1950-1986 | | 1.13 | (0.54, 2.07) | 2.9 |
| Schlicht 1990, women | 13 | 77% | Australia | 1950-1986 | | 5.01 | (1.01, 14.65) | 0.2 |
| Shang 2011, men and women | 23 | - | Asia | 1990-2006 | | 0.14 | (0.09, 0.21) | 3.6 |
| **Sub-total** (I2= 79.5%, p= 0.002) | | | | | | **0.79** | **(0.03, 1.62)** | **6.7** |
| **Africa** | | | | | | | | |
| Dean 1969, men | 23 | 96% | Africa | 1960-1966 | | 1.26 | (0.74, 2.13) | 3.0 |
| **Sub-total** (I2= .%, p= .) | | | | | | **1.26** | **(0.56, 1.96)** | **3.0** |
| **Overall** (I2= 93.9%, p< 0.001 ) | | | | | | **1.44** | **(1.16, 1.72)** | **100** |

**Fig 6. Meta-analysis of standardized mortality rate for suicides by geographic zones.**

$P < 0.001$), than other medical specialties (0.24; 95CI 0.18, 0.30; $P < 0.001$), than surgeons (0.25; 95CI 0.19, 0.30; $P < 0.001$) and then internships (0.24; 95CI 0.15, 0.34; $P < 0.001$). Moreover, a higher risk of suicide in internal medicine than in other medical specialties (0.12; 95CI 0.08, 0.17; $P < 0.001$), than surgeons (0.13; 95CI 0.08, 0.18; $P < 0.001$), and than internships (0.13; 95CI 0.03, 0.22; $P = 0.008$). Finally, we demonstrated a higher risk of suicide in psychiatrists than other medical specialties (0.07; 95CI 0.02, 0.13; $P = 0.009$) and than surgeons (0.08; 95CI 0.02, 0.13; $P = 0.005$) (S1 Fig).

## Meta-analysis of percentages of suicide in physicians by category of surgical specialties

We included six studies [15,16,43,47,51,83]. The percentage of suicide in general surgeons was 6% i.e. (95CI 4, 9; $I^2 = 64.5\%$, $P = 0.04$), in obstetricians was 4% (95CI 2, 5; $I^2 = 0$, $P =$

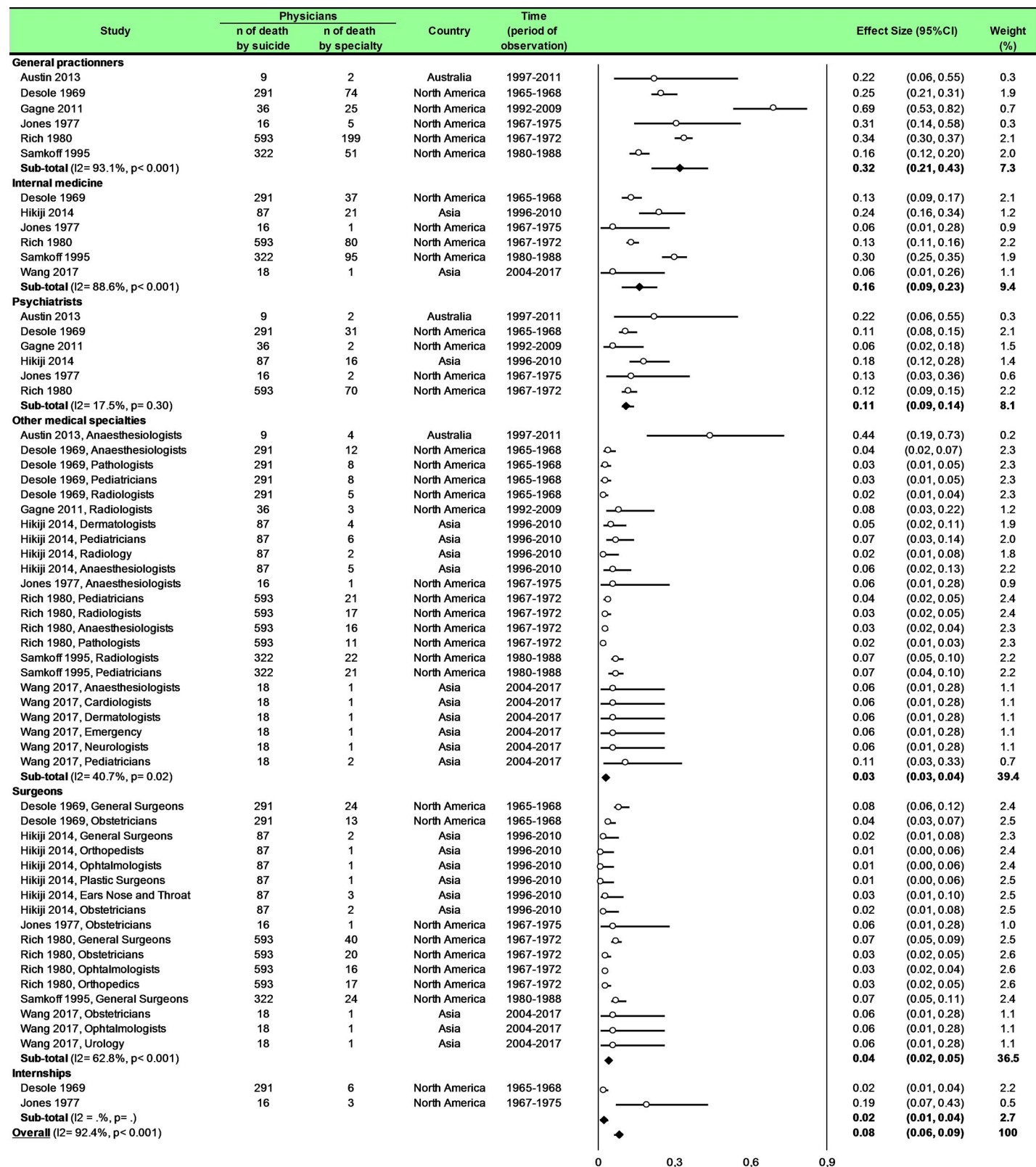

| Study | Physicians n of death by suicide | Physicians n of death by specialty | Country | Time (period of observation) | | Effect Size (95%CI) | | Weight (%) |
|---|---|---|---|---|---|---|---|---|
| **General practionners** | | | | | | | | |
| Austin 2013 | 9 | 2 | Australia | 1997-2011 | | 0.22 | (0.06, 0.55) | 0.3 |
| Desole 1969 | 291 | 74 | North America | 1965-1968 | | 0.25 | (0.21, 0.31) | 1.9 |
| Gagne 2011 | 36 | 25 | North America | 1992-2009 | | 0.69 | (0.53, 0.82) | 0.7 |
| Jones 1977 | 16 | 5 | North America | 1967-1975 | | 0.31 | (0.14, 0.58) | 0.3 |
| Rich 1980 | 593 | 199 | North America | 1967-1972 | | 0.34 | (0.30, 0.37) | 2.1 |
| Samkoff 1995 | 322 | 51 | North America | 1980-1988 | | 0.16 | (0.12, 0.20) | 2.0 |
| Sub-total (I2= 93.1%, p< 0.001) | | | | | | **0.32** | **(0.21, 0.43)** | **7.3** |
| **Internal medicine** | | | | | | | | |
| Desole 1969 | 291 | 37 | North America | 1965-1968 | | 0.13 | (0.09, 0.17) | 2.1 |
| Hikiji 2014 | 87 | 21 | Asia | 1996-2010 | | 0.24 | (0.16, 0.34) | 1.2 |
| Jones 1977 | 16 | 1 | North America | 1967-1975 | | 0.06 | (0.01, 0.28) | 0.9 |
| Rich 1980 | 593 | 80 | North America | 1967-1972 | | 0.13 | (0.11, 0.16) | 2.2 |
| Samkoff 1995 | 322 | 95 | North America | 1980-1988 | | 0.30 | (0.25, 0.35) | 1.9 |
| Wang 2017 | 18 | 1 | Asia | 2004-2017 | | 0.06 | (0.01, 0.26) | 1.1 |
| Sub-total (I2= 88.6%, p< 0.001) | | | | | | **0.16** | **(0.09, 0.23)** | **9.4** |
| **Psychiatrists** | | | | | | | | |
| Austin 2013 | 9 | 2 | Australia | 1997-2011 | | 0.22 | (0.06, 0.55) | 0.3 |
| Desole 1969 | 291 | 31 | North America | 1965-1968 | | 0.11 | (0.08, 0.15) | 2.1 |
| Gagne 2011 | 36 | 2 | North America | 1992-2009 | | 0.06 | (0.02, 0.18) | 1.5 |
| Hikiji 2014 | 87 | 16 | Asia | 1996-2010 | | 0.18 | (0.12, 0.28) | 1.4 |
| Jones 1977 | 16 | 2 | North America | 1967-1975 | | 0.13 | (0.03, 0.36) | 0.6 |
| Rich 1980 | 593 | 70 | North America | 1967-1972 | | 0.12 | (0.09, 0.15) | 2.2 |
| Sub-total (I2= 17.5%, p= 0.30) | | | | | | **0.11** | **(0.09, 0.14)** | **8.1** |
| **Other medical specialties** | | | | | | | | |
| Austin 2013, Anaesthesiologists | 9 | 4 | Australia | 1997-2011 | | 0.44 | (0.19, 0.73) | 0.2 |
| Desole 1969, Anaesthesiologists | 291 | 12 | North America | 1965-1968 | | 0.04 | (0.02, 0.07) | 2.3 |
| Desole 1969, Pathologists | 291 | 8 | North America | 1965-1968 | | 0.03 | (0.01, 0.05) | 2.3 |
| Desole 1969, Pediatricians | 291 | 8 | North America | 1965-1968 | | 0.03 | (0.01, 0.05) | 2.3 |
| Desole 1969, Radiologists | 291 | 5 | North America | 1965-1968 | | 0.02 | (0.01, 0.04) | 2.3 |
| Gagne 2011, Radiologists | 36 | 3 | North America | 1992-2009 | | 0.08 | (0.03, 0.22) | 1.2 |
| Hikiji 2014, Dermatologists | 87 | 4 | Asia | 1996-2010 | | 0.05 | (0.02, 0.11) | 1.9 |
| Hikiji 2014, Pediatricians | 87 | 6 | Asia | 1996-2010 | | 0.07 | (0.03, 0.14) | 2.0 |
| Hikiji 2014, Radiology | 87 | 2 | Asia | 1996-2010 | | 0.02 | (0.01, 0.08) | 1.8 |
| Hikiji 2014, Anaesthesiologists | 87 | 5 | Asia | 1996-2010 | | 0.06 | (0.02, 0.13) | 2.2 |
| Jones 1977, Anaesthesiologists | 16 | 1 | North America | 1967-1975 | | 0.06 | (0.01, 0.28) | 0.9 |
| Rich 1980, Pediatricians | 593 | 21 | North America | 1967-1972 | | 0.04 | (0.02, 0.05) | 2.4 |
| Rich 1980, Radiologists | 593 | 17 | North America | 1967-1972 | | 0.03 | (0.02, 0.05) | 2.4 |
| Rich 1980, Anaesthesiologists | 593 | 16 | North America | 1967-1972 | | 0.03 | (0.02, 0.04) | 2.3 |
| Rich 1980, Pathologists | 593 | 11 | North America | 1967-1972 | | 0.02 | (0.01, 0.03) | 2.3 |
| Samkoff 1995, Radiologists | 322 | 22 | North America | 1980-1988 | | 0.07 | (0.05, 0.10) | 2.2 |
| Samkoff 1995, Pediatricians | 322 | 21 | North America | 1980-1988 | | 0.07 | (0.04, 0.10) | 2.2 |
| Wang 2017, Anaesthesiologists | 18 | 1 | Asia | 2004-2017 | | 0.06 | (0.01, 0.28) | 1.1 |
| Wang 2017, Cardiologists | 18 | 1 | Asia | 2004-2017 | | 0.06 | (0.01, 0.28) | 1.1 |
| Wang 2017, Dermatologists | 18 | 1 | Asia | 2004-2017 | | 0.06 | (0.01, 0.28) | 1.1 |
| Wang 2017, Emergency | 18 | 1 | Asia | 2004-2017 | | 0.06 | (0.01, 0.28) | 1.1 |
| Wang 2017, Neurologists | 18 | 1 | Asia | 2004-2017 | | 0.06 | (0.01, 0.28) | 1.1 |
| Wang 2017, Pediatricians | 18 | 2 | Asia | 2004-2017 | | 0.11 | (0.03, 0.33) | 0.7 |
| Sub-total (I2= 40.7%, p= 0.02) | | | | | | **0.03** | **(0.03, 0.04)** | **39.4** |
| **Surgeons** | | | | | | | | |
| Desole 1969, General Surgeons | 291 | 24 | North America | 1965-1968 | | 0.08 | (0.06, 0.12) | 2.4 |
| Desole 1969, Obstetricians | 291 | 13 | North America | 1965-1968 | | 0.04 | (0.03, 0.07) | 2.5 |
| Hikiji 2014, General Surgeons | 87 | 2 | Asia | 1996-2010 | | 0.02 | (0.01, 0.08) | 2.3 |
| Hikiji 2014, Orthopedists | 87 | 1 | Asia | 1996-2010 | | 0.01 | (0.00, 0.06) | 2.4 |
| Hikiji 2014, Ophtalmologists | 87 | 1 | Asia | 1996-2010 | | 0.01 | (0.00, 0.06) | 2.4 |
| Hikiji 2014, Plastic Surgeons | 87 | 1 | Asia | 1996-2010 | | 0.01 | (0.00, 0.06) | 2.5 |
| Hikiji 2014, Ears Nose and Throat | 87 | 3 | Asia | 1996-2010 | | 0.03 | (0.01, 0.10) | 2.5 |
| Hikiji 2014, Obstetricians | 87 | 2 | Asia | 1996-2010 | | 0.02 | (0.01, 0.08) | 2.5 |
| Jones 1977, Obstetricians | 16 | 1 | North America | 1967-1975 | | 0.06 | (0.01, 0.28) | 1.0 |
| Rich 1980, General Surgeons | 593 | 40 | North America | 1967-1972 | | 0.07 | (0.05, 0.09) | 2.5 |
| Rich 1980, Obstetricians | 593 | 20 | North America | 1967-1972 | | 0.03 | (0.02, 0.05) | 2.6 |
| Rich 1980, Ophtalmologists | 593 | 16 | North America | 1967-1972 | | 0.03 | (0.02, 0.04) | 2.6 |
| Rich 1980, Orthopedics | 593 | 17 | North America | 1967-1972 | | 0.03 | (0.02, 0.05) | 2.6 |
| Samkoff 1995, General Surgeons | 322 | 24 | North America | 1980-1988 | | 0.07 | (0.05, 0.11) | 2.4 |
| Wang 2017, Obstetricians | 18 | 1 | Asia | 2004-2017 | | 0.06 | (0.01, 0.28) | 1.1 |
| Wang 2017, Ophtalmologists | 18 | 1 | Asia | 2004-2017 | | 0.06 | (0.01, 0.28) | 1.1 |
| Wang 2017, Urology | 18 | 1 | Asia | 2004-2017 | | 0.06 | (0.01, 0.28) | 1.1 |
| Sub-total (I2= 62.8%, p< 0.001) | | | | | | **0.04** | **(0.02, 0.05)** | **36.5** |
| **Internships** | | | | | | | | |
| Desole 1969 | 291 | 6 | North America | 1965-1968 | | 0.02 | (0.01, 0.04) | 2.2 |
| Jones 1977 | 16 | 3 | North America | 1967-1975 | | 0.19 | (0.07, 0.43) | 0.5 |
| Sub-total (I2 = .%, p= .) | | | | | | **0.02** | **(0.01, 0.04)** | **2.7** |
| <u>Overall</u> (I2= 92.4%, p< 0.001) | | | | | | **0.08** | **(0.06, 0.09)** | **100** |

0    0,3    0,6    0,9

**Fig 7. Meta-analysis of percentages of suicide in physicians by group of specialties.**

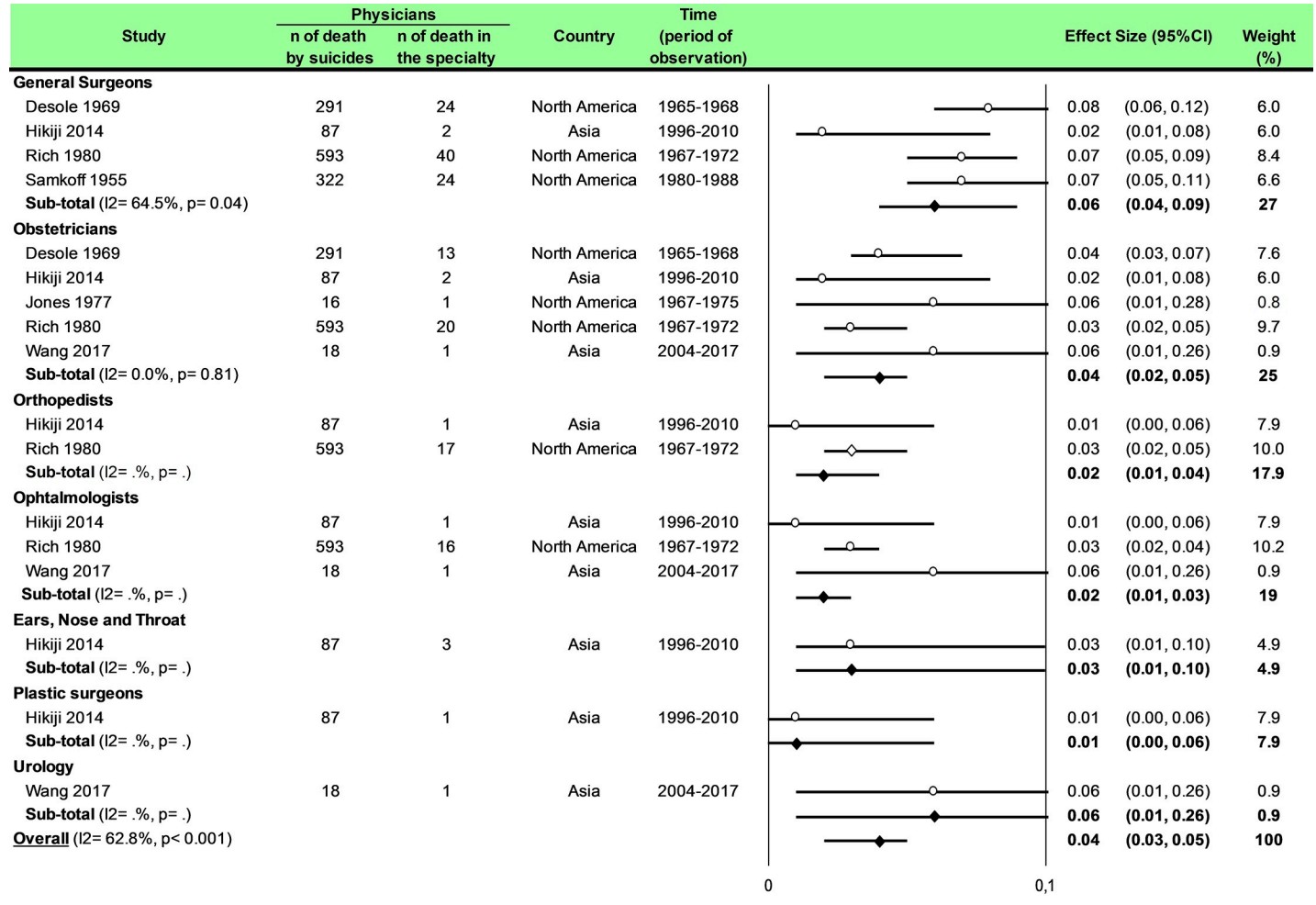

**Fig 8. Meta-analysis of percentages of suicide in physicians by category of surgical specialties.**

0.81*)*, in orthopaedists was 2% (95CI 1, 4), in ears, nose and throat was 3% (95CI 0, 3) and in plastic surgeons was 1% (95CI 0, 6) (Fig 8).

Meta-regressions demonstrated a higher risk of suicide in general surgeons than obstetricians (0.03; 95CI 0.01, 0.05; *P* = 0.035), than orthopedists (0.04; 95CI 0.01, 0.07; *P* = 0.006), than ophthalmologists (0.04; 95CI 0.02, 0.07; *P* = 0.006) and than plastic surgeons (0.05; 95CI 0.01, 0.09; *P* = 0.010) (S2 Fig).

## Meta-analysis of percentages of suicide in physicians by category of medical specialties

Eight studies were included [15,16,40,43,45,47,51,83]. The percentage of suicide in internal medicine was 16% (95CI 9, 23; $I^2$ = 88.6%, *P* < 0.001), in psychiatrists was 11% (95CI 9, 14; $I^2$ = 17.5%, *P* = 0.30), in anaesthesiologists was 4% (95CI 2, 6; $I^2$ = 43.6%, *P* = 0.11), in radiologists was 3% (95CI 2, 5; $I^2$ = 66.0%, *P* = 0.02), in paediatricians was 4% (95CI 3, 6; $I^2$ = 46.4%, *P* = 0.11), in pathologists was 2% (95CI 1, 3), in dermatologists was 5% (95CI 1, 9), in cardiologists was 6% (95CI 1, 26), in neurologists was 6% (95CI 1, 26) and in emergency physicians was 6% (95CI 1, 26) (Fig 9). Meta-regressions demonstrated a higher risk of suicide in internal medicine than anesthesiologists (0.12; 95CI 0.06, 0.18; *P* = 0.001) than radiologists (0.13; 95CI

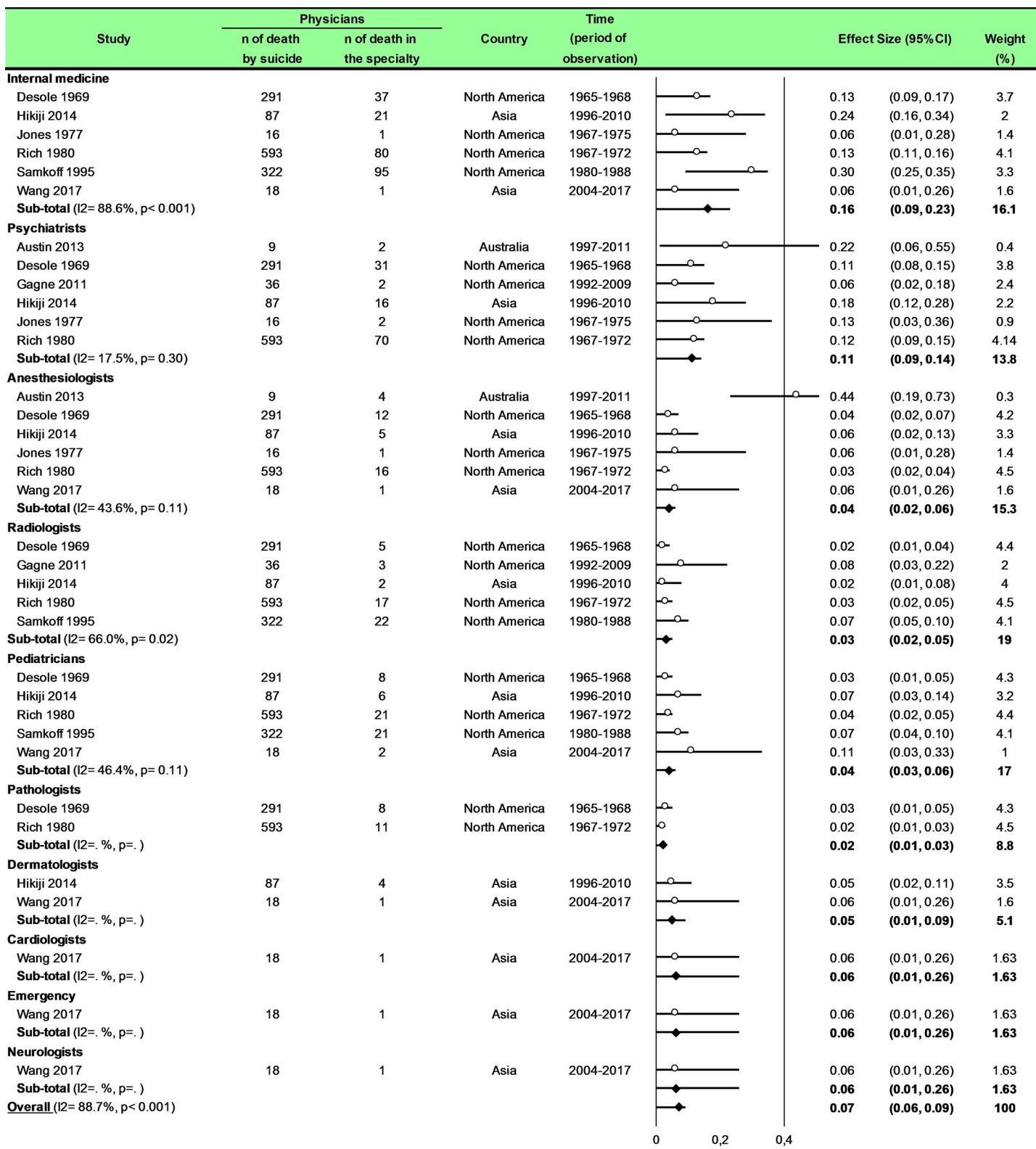

| Study | Physicians | | Country | Time (period of observation) | Effect Size (95%CI) | | Weight (%) |
|---|---|---|---|---|---|---|---|
| | n of death by suicide | n of death in the specialty | | | | | |
| **Internal medicine** | | | | | | | |
| Desole 1969 | 291 | 37 | North America | 1965-1968 | 0.13 | (0.09, 0.17) | 3.7 |
| Hikiji 2014 | 87 | 21 | Asia | 1996-2010 | 0.24 | (0.16, 0.34) | 2 |
| Jones 1977 | 16 | 1 | North America | 1967-1975 | 0.06 | (0.01, 0.28) | 1.4 |
| Rich 1980 | 593 | 80 | North America | 1967-1972 | 0.13 | (0.11, 0.16) | 4.1 |
| Samkoff 1995 | 322 | 95 | North America | 1980-1988 | 0.30 | (0.25, 0.35) | 3.3 |
| Wang 2017 | 18 | 1 | Asia | 2004-2017 | 0.06 | (0.01, 0.26) | 1.6 |
| **Sub-total** (I2= 88.6%, p< 0.001) | | | | | **0.16** | **(0.09, 0.23)** | **16.1** |
| **Psychiatrists** | | | | | | | |
| Austin 2013 | 9 | 2 | Australia | 1997-2011 | 0.22 | (0.06, 0.55) | 0.4 |
| Desole 1969 | 291 | 31 | North America | 1965-1968 | 0.11 | (0.08, 0.15) | 3.8 |
| Gagne 2011 | 36 | 2 | North America | 1992-2009 | 0.06 | (0.02, 0.18) | 2.4 |
| Hikiji 2014 | 87 | 16 | Asia | 1996-2010 | 0.18 | (0.12, 0.28) | 2.2 |
| Jones 1977 | 16 | 2 | North America | 1967-1975 | 0.13 | (0.03, 0.36) | 0.9 |
| Rich 1980 | 593 | 70 | North America | 1967-1972 | 0.12 | (0.09, 0.15) | 4.14 |
| **Sub-total** (I2= 17.5%, p= 0.30) | | | | | **0.11** | **(0.09, 0.14)** | **13.8** |
| **Anesthesiologists** | | | | | | | |
| Austin 2013 | 9 | 4 | Australia | 1997-2011 | 0.44 | (0.19, 0.73) | 0.3 |
| Desole 1969 | 291 | 12 | North America | 1965-1968 | 0.04 | (0.02, 0.07) | 4.2 |
| Hikiji 2014 | 87 | 5 | Asia | 1996-2010 | 0.06 | (0.02, 0.13) | 3.3 |
| Jones 1977 | 16 | 1 | North America | 1967-1975 | 0.06 | (0.01, 0.28) | 1.4 |
| Rich 1980 | 593 | 16 | North America | 1967-1972 | 0.03 | (0.02, 0.04) | 4.5 |
| Wang 2017 | 18 | 1 | Asia | 2004-2017 | 0.06 | (0.01, 0.26) | 1.6 |
| **Sub-total** (I2= 43.6%, p= 0.11) | | | | | **0.04** | **(0.02, 0.06)** | **15.3** |
| **Radiologists** | | | | | | | |
| Desole 1969 | 291 | 5 | North America | 1965-1968 | 0.02 | (0.01, 0.04) | 4.4 |
| Gagne 2011 | 36 | 3 | North America | 1992-2009 | 0.08 | (0.03, 0.22) | 2 |
| Hikiji 2014 | 87 | 2 | Asia | 1996-2010 | 0.02 | (0.01, 0.08) | 4 |
| Rich 1980 | 593 | 17 | North America | 1967-1972 | 0.03 | (0.02, 0.05) | 4.5 |
| Samkoff 1995 | 322 | 22 | North America | 1980-1988 | 0.07 | (0.05, 0.10) | 4.1 |
| **Sub-total** (I2= 66.0%, p= 0.02) | | | | | **0.03** | **(0.02, 0.05)** | **19** |
| **Pediatricians** | | | | | | | |
| Desole 1969 | 291 | 8 | North America | 1965-1968 | 0.03 | (0.01, 0.05) | 4.3 |
| Hikiji 2014 | 87 | 6 | Asia | 1996-2010 | 0.07 | (0.03, 0.14) | 3.2 |
| Rich 1980 | 593 | 21 | North America | 1967-1972 | 0.04 | (0.02, 0.05) | 4.4 |
| Samkoff 1995 | 322 | 21 | North America | 1980-1988 | 0.07 | (0.04, 0.10) | 4.1 |
| Wang 2017 | 18 | 2 | Asia | 2004-2017 | 0.11 | (0.03, 0.33) | 1 |
| **Sub-total** (I2= 46.4%, p= 0.11) | | | | | **0.04** | **(0.03, 0.06)** | **17** |
| **Pathologists** | | | | | | | |
| Desole 1969 | 291 | 8 | North America | 1965-1968 | 0.03 | (0.01, 0.05) | 4.3 |
| Rich 1980 | 593 | 11 | North America | 1967-1972 | 0.02 | (0.01, 0.03) | 4.5 |
| **Sub-total** (I2=. %, p=. ) | | | | | **0.02** | **(0.01, 0.03)** | **8.8** |
| **Dermatologists** | | | | | | | |
| Hikiji 2014 | 87 | 4 | Asia | 1996-2010 | 0.05 | (0.02, 0.11) | 3.5 |
| Wang 2017 | 18 | 1 | Asia | 2004-2017 | 0.06 | (0.01, 0.26) | 1.6 |
| **Sub-total** (I2=. %, p=. ) | | | | | **0.05** | **(0.01, 0.09)** | **5.1** |
| **Cardiologists** | | | | | | | |
| Wang 2017 | 18 | 1 | Asia | 2004-2017 | 0.06 | (0.01, 0.26) | 1.63 |
| **Sub-total** (I2=. %, p=. ) | | | | | **0.06** | **(0.01, 0.26)** | **1.63** |
| **Emergency** | | | | | | | |
| Wang 2017 | 18 | 1 | Asia | 2004-2017 | 0.06 | (0.01, 0.26) | 1.63 |
| **Sub-total** (I2=. %, p=. ) | | | | | **0.06** | **(0.01, 0.26)** | **1.63** |
| **Neurologists** | | | | | | | |
| Wang 2017 | 18 | 1 | Asia | 2004-2017 | 0.06 | (0.01, 0.26) | 1.63 |
| **Sub-total** (I2=. %, p=. ) | | | | | **0.06** | **(0.01, 0.26)** | **1.63** |
| **Overall** (I2= 88.7%, p< 0.001) | | | | | **0.07** | **(0.06, 0.09)** | **100** |

**Fig 9. Meta-analysis of percentages of suicide in physicians by category of medical specialties.**

| Study | Physicians | | | | Country | Time (period of observation) | Effect Size (95%CI) | | Weight (%) |
|---|---|---|---|---|---|---|---|---|---|
| | n total of death | Sex (%male) | n death by suicides | Sex (%male) | | | | | |
| No Author 1986 | 1 258 | - | 24 | 37% | North America | 1980-1981 | 0.02 | (0.01, 0.03) | 10.1 |
| Carpenter 1997 | 2 798 | 94% | 65 | 87% | Europe | 1962-1979 | 0.02 | (0.02, 0.03) | 10.6 |
| Craig 1968 | 8 372 | 96% | 228 | 92% | North America | 1965-1967 | 0.03 | (0.02, 0.03) | 11.0 |
| Everson 1975 | 493 | - | 88 | - | North America | 1966-1970 | 0.18 | (0.15, 0.21) | 3.4 |
| Innos 2002, men | 160 | 100% | 6 | 100% | Europe | 1983-1998 | 0.07 | (0.04, 0.12) | 2.7 |
| Innos 2002, women | 195 | 0% | 5 | 0% | Europe | 1983-1998 | 0.06 | (0.03, 0.10) | 3.6 |
| Lew 1979 | 637 | - | 38 | - | North America | 1954-1976 | 0.06 | (0.04, 0.08) | 6.7 |
| Linde 1981 | 274 | 100% | 10 | 100% | North America | 1930-1946 | 0.04 | (0.02, 0.07) | 5.6 |
| Palhares-Alves 2015 | 2 297 | 87% | 50 | 76% | South America | 2000-2009 | 0.02 | (0.02, 0.03) | 10.5 |
| Rich 1980 | 18 730 | 96% | 593 | 92% | North America | 1967-1972 | 0.03 | (0.03, 0.03) | 11.1 |
| Samkoff 1995 | 835 | - | 32 | - | North America | 1980-1988 | 0.04 | (0.03, 0.05) | 8.4 |
| Schlicht 1990 | 126 | 91% | 13 | 77% | Australia | 1950-1986 | 0.10 | (0.06, 0.17) | 1.68 |
| Shang 2012, surgeons | 161 | - | 3 | - | Asia | 1990-2006 | 0.02 | (0.01, 0.05) | 6.0 |
| Shang 2012, anaesthesiologists | 16 | - | 1 | - | Asia | 1990-2006 | 0.06 | (0.01, 0.28) | 0.4 |
| **Overall** (I2= 88.7%, p< 0.001) | | | | | | | **0.04** | **(0.03, 0.05)** | **100** |

**Fig 10. Meta-analysis of prevalence of physicians died by suicide among all deaths in physicians.**

0.07, 0.19; $P < 0.001$), than pediatricians (0.12; 95CI 0.06, 0.18; $P = 0.001$) than pathologists (0.14; 95CI 0.07, 0.21; $P < 0.001$) and than dermatologists (0.12; 95CI 0.03, 0.21; $P = 0.13$). Moreover, the risk of suicide was higher in psychiatrists than anesthesiologists (0.07; 95CI 0.01, 0.13; $P = 0.038$), than radiologists (0.08; 95CI 0.02, 0.14; $P = 0.014$), than pediatricians (0.07; 95CI 0.01, 0.13; $P = 0.038$) and than pathologists (0.09; 95CI 0.02, 0.17; $P = 0.014$) (S3 Fig).

## Meta-analysis of prevalence of physicians dead by suicide among all deaths in physicians

We included 12 studies [16,39,41,42,44,46,48,49,50,51,52,53], and we demonstrated a prevalence of 4% (95CI 3, 5) with an important heterogeneity ($I^2 = 88.7\%$) (Fig 10).

Meta-regression on geographic zones did not retrieves any significant result. Moreover, insufficient data did not permit other meta-regression.

| Study | Physicians | | | | Country | Time (period of observation) | Effect Size (95%CI) | | Weight (%) |
|---|---|---|---|---|---|---|---|---|---|
| | n total alive | Sex (%male) | n with suicidal ideation | Sex (%male) | | | | | |
| **All** | | | | | | | | | |
| Lindfors 2009 | 328 | 53% | 73 | - | Europe | 2004-2008 | 0.22 | (0.18, 0.27) | 8.4 |
| Simon 1968 | 36 | - | 13 | - | North America | - | 0.36 | (0.22, 0.52) | 4.6 |
| Sub-total (I2=.%, p=.) | | | | | | | **0.23** | **(0.19, 0.28)** | **13.0** |
| **Men** | | | | | | | | | |
| Brooks 2017 | 1188 | 72% | 38 | 54% | North America | 2003-2014 | 0.02 | (0.02, 0.03) | 9.1 |
| Hem 2000 | 1063 | 70% | 1004 | 72% | Europe | 1993-1999 | 0.08 | (0.07, 0.11) | 8.9 |
| Loas 2018 | 223 | 40% | 42 | 19% | Europe | 2015-2018 | 0.08 | (0.06, 0.10) | 8.9 |
| Olkinuora 1990 | 2644 | 57% | 609 | 56% | Europe | 1986-1989 | 0.21 | (0.20, 0.24) | 8.9 |
| Sub-total (I2=99.1%, p< 0.001) | | | | | | | **0.10** | **(0.01, 0.18)** | **35.8** |
| **Women** | | | | | | | | | |
| Brooks 2017 | 1188 | 72% | 32 | 54% | North America | 2003-2014 | 0.02 | (0.01, 0.03) | 9.1 |
| Fridner 2009, Italy group | 126 | 0% | 27 | 0% | Europe | 2005-2005 | 0.21 | (0.15, 0.29) | 7.6 |
| Fridner 2009, Sweden group | 385 | 0% | 122 | 0% | Europe | 2005-2005 | 0.32 | (0.27, 0.36) | 8.4 |
| Hem 2000 | 1063 | 70% | 1004 | 72% | Europe | 1993-1999 | 0.15 | (0.12, 0.20) | 8.5 |
| Loas 2018 | 223 | 40% | 91 | 19% | Europe | 2015-2018 | 0.16 | (0.13, 0.20) | 8.8 |
| Olkinuora 1990 | 2644 | 57% | 609 | 56% | Europe | 1986-1989 | 0.25 | (0.23, 0.28) | 8.9 |
| Sub-total (I2=99.1%, p< 0.001) | | | | | | | **0.19** | **(0.07, 0.30)** | **51.3** |
| **Overall** (I2=98.8%, p< 0.001) | | | | | | | **0.17** | **(0.12, 0.21)** | **100** |

**Fig 11. Meta-analysis of prevalence of physicians with suicidal ideation among all the physicians.**

**Table 1. Characteristics of included studies.** CI, Confidence Interval; n, Number; SMR, Standardized Mortality Ratio; USA, United States of America.

| Study | Country | Continent | Time Period | Total Physicians–n (%) | | Suicides Death–n (%) | | Mortality–SMR (95CI) | | Attempts—n | | Thoughts—n | | Specialities |
|-------|---------|-----------|-------------|------|------|------|------|------|------|------|------|------|------|--------------|
| | | | | Men | Women | Men | Women | Men | Women | Men | Women | Men | Women | |
| Aasland 2001 | Norway | Europe | 1960–1993 | | | 73 (89) | 9 (11) | | | | | | | No specified |
| Aasland 2011 | Norway | Europe | 1960–2000 | | | | | | | | | | | No specified |
| Arnetz 1987 | Sweden | Europe | 1961–1970 | | | 32 (76) | 10 (24) | 1,2 (0.85, 1.69) | 5,7 (1.68, 10.7) | | | | | No specified |
| Austin 2013 | Australia | Australia, New-Zealand and Pacific | 1997–2011 | | | 6 (66) | 3 (34) | | | | | | | Anaesthesiologists, psychiatrists, general practitioners, general surgeons |
| Bamayr 1986 | Germany | Europe | 1963–1978 | | | 67 (71) | 27 (29) | 1,58 (1.07, 2.34) | 2,96 (1.44, 6.09) | | | | | No specified |
| Brooks 2017 | USA | North America | 2003–2014 | 1188 (72) | 544 (28) | | | | | | | 38 | 32 | No specified |
| Carpenter 1997 | Great Britain | Europe | 1962–1979 | | | 56 (87) | 8 (13) | 0,96 (0.72, 1.25) | 2,15 (0.93, 4.23) | | | | | No specified |
| Craig 1968 | USA | North America | 1965–1967 | | | 211 | 17 | | | | | | | No specified |
| Davidson 2018 | USA | North America | 2005–2015 | | | | | 2,29 (1.66, 3.08) | 2,29 (1.66, 3.08) | | | | | No specified |
| Dean 1969 | South Africa | Africa | 1960–1966 | | | 22 (96) | 1 (4) | 1,26 (0.74, 2.13) | | | | | | No specified |
| Desole 1969 | USA | North America | 1965–1968 | | | | | | | | | | | General practitioners, general surgeons, internal medicine, psychiatrists, obstetricians, anaesthesiologists, pathology, paediatrics, radiology, internships |
| Everson 1975 | USA | North America | 1966–1970 | | | | | | | | | | | No specified |
| Frank 1999 | USA | North America | 1993–1994 | 0 | 4501 (100) | | | | | | 61 | | | No specified |
| Frank 2000 | USA | North America | 1984–1995 | | | 379 (91) | 37 (9) | 1,7 (1.53, 1.88) | 2,38 (1.69, 3.28) | | | | | No specified |
| Fridner 2009 | Sweden and Italy | Europe | 2005–2005 | 0 | 385 (100) | | | | | | | | 122 | No specified |
| Gagne 2011 | Quebec | North America | 1992–2009 | | | 29 (80) | 7 (20) | | | | | | | General practitioners, radiology, psychiatrists |
| Gold 2013 | USA | North America | 2003–2008 | | | | | | | | | | | No specified |
| Gunnarsdottir 1995 | Iceland | Europe | 1920–1979 | | | | | | | | | | | No specified |
| Hawton 2001 | Great Britain | Europe | 1991–1995 | | | 42 (74) | 15 (26) | 0,67 (0.47, 0.87) | 2,02 (1.00, 3.04) | | | | | No specified |

*(Continued)*

**Table 1.** (Continued)

| Study | Country | Continent | Time Period | Total Physicians–n (%) | | Suicides | | | | | | | | Specialities |
|-------|---------|-----------|-------------|-----------|-------|-----------|-----------|------------|------------|---------|-----------|---------|-----------|-------------|
| | | | | | | Death–n (%) | | Mortality–SMR (95CI) | | Attempts—n | | Thoughts—n | | |
| | | | | Men | Women | Men | Women | Men | Women | Men | Women | Men | Women | |
| Hawton 2002 | England and Wales | Europe | 1994–1997 | | | | | | | | | | | No specified |
| Hawton 2011 | Danish | Europe | 1981–2006 | | | 131 (80) | 32 (20) | | | | | | | No specified |
| Hem 2000 | Norway | Europe | 1993–1999 | 722 (72) | 282 (28) | | | | | 7 | 9 | 61 | 43 | No specified |
| Hem 2005 | Norway | Europe | 1960–1990 | | | 98 (88) | 13 (22) | | | | | | | No specified |
| Hemenway 1993 | USA | North America | 1976–1988 | | | | | | | | | | | No specified |
| Herner 1993 | Sweden | Europe | 1989–1991 | | | 17 (68) | 8 (32) | 1.1 (0.8, 1.52) | 2,32 (1.12, 4.81) | | | | | No specified |
| Hikiji 2014 | Japan | Asia | 1996–2010 | | | 68 (79) | 19 (21) | | | | | | | Internal medicine, dermatologists, paediatrics, psychiatrists, general surgeons, orthopaedists, ophthalmology, plastic surgeons, ENT, obstetricians, radiology, anaesthesiologists |
| Hubbard 1922 | USA | North America | 1921 | | | | | | | | | | | No specified |
| Innos 2002 | Estonia | Europe | 1983–1998 | | | 6 (54) | 5 (46) | 0,58 (0.21, 1.27) | 0,62 (0.20, 1.45) | | | | | No specified |
| Jones 1977 | USA | North America | 1967–1975 | | | | | | | 11 | 5 | | | General practitioners, anaesthesiologists, internal medicine, obstetricians, psychiatrists, general surgeons, internships |
| Juel 1999 | Danish | Europe | 1973–1992 | | | 168 (86) | 26 (14) | 1.64 (1.40, 1.91) | 1.68 (1.10, 2.46) | | | | | No specified |
| Lew 1976 | USA | North America | 1954–1976 | | | | | | | | | | | No specified |
| Linde 1981 | USA | North America | 1930–1946 | 274 (100) | 0 | 10 | 0 | | | | | | | No specified |
| Lindeman 1997 | Finland | Europe | 1986–1993 | | | | | | | | | | | No specified |
| Lindeman 2007 | Finland | Europe | 1987–1988 | | | 2 (28) | 5 (72) | | | | | | | No specified |
| Lindfors 2009 | Finland | Europe | 2004–2008 | 175 (53) | 153 (47) | | | | | | | | | No specified |
| Lindhardt 1963 | Denmark | Europe | 1935–1959 | | | | | 1.53 (1.06, 2.20) | | | | | | No specified |
| Loas 2018 | Belgium | Europe | 2015–2018 | 223 (40) | 334 (60) | | | | | 5 | 9 | 42 | 91 | No specified |
| No Author 1986 | USA | North America | 1980–1981 | | | | | | | | | | | No specified |

*(Continued)*

**Table 1.** (Continued)

| Study | Country | Continent | Time Period | Total Physicians–n (%) | | Suicides Death–n (%) | | Mortality–SMR (95CI) | | Attempts—n | | Thoughts—n | | Specialities |
|---|---|---|---|---|---|---|---|---|---|---|---|---|---|---|
| | | | | Men | Women | Men | Women | Men | Women | Men | Women | Men | Women | |
| Nordentoft 1988 | Netherlands | Europe | 1970–1980 | | | 59 (85) | 10 (15) | 2.46 (1.02, 3.42) | 3.33 (0.42, 26.3) | | | | | No specified |
| Olkinuora 1990 | Finland | Europe | 1986–1989 | 1582 (59) | 1062 (41) | | | | | 10 | 6 | 340 | 269 | No specified |
| Palhares-Alves 2015 | Brazil | South America | 2000–2009 | | | 38 (76) | 12 (24) | | | | | | | No specified |
| Petersen 2008 | USA | North America | 1984–1992 | | | 181 (89) | 22 (11) | 0.8 (0.53, 1.20) | 2.39 (1.52, 3.77) | | | | | No specified |
| Pitts 1979 | USA | North America | 1967–1972 | | 751 | | 49 | | 3.57 (1.23, 10.4) | | | | | No specified |
| Rafnsson 1998 | Island | Europe | 1955–1995 | | | 7 (100) | | 1.01 (0.40, 2.04) | | | | | | No specified |
| Revicki 1985 | USA | North America | 1978–1982 | | | 13 | | 1.16 (0.80, 1.70) | | | | | | No specified |
| Rich 1979 | USA | North America | 1967–1972 | 17979 | | 544 | | 1.03 (0.74, 1.45) | | | | | | No specified |
| Rich 1980 | USA | North America | 1967–1972 | | | 544 (92) | 49 (8) | | | | | | | General practitioners, internal medicine, general surgeons, psychiatrists, obstetricians, paediatrics, radiology, anaesthesiologists, pathology, ophthalmology, orthopaedists |
| Rimpela 1987 | Finland | Europe | 1971–1980 | | | 17 | | 1.28 (1.01, 1.65) | | | | | | No specified |
| Rose 1973 | USA | North America | 1959–1961 | | | 48 (98) | 1 (2) | 2.03 (1.29, 3.19) | | | | | | No specified |
| Roy 1985 | USA | North America | 1981–1974 | | | | | | | | | | | No specified |
| Samkoff 1995 | USA | North America | 1980–1988 | | | | | | | | | | | General practitioners, internal medicine, general surgeons, radiology, paediatrics |
| Schlicht 1990 | Australia | Australia, New-Zealand and Pacific | 1950–1986 | 1279 (88) | 174 (12) | 10 | 3 | 1.13 (0.54, 2.07) | 5.01 (1.01, 14.7) | | | | | No specified |
| Shang 2011 | Taiwan | Australia, New-Zealand and Pacific | 1990–2006 | | | | | | | | | | | No specified |

(Continued)

**Table 1.** (Continued)

| Study | Country | Continent | Time Period | Total Physicians–n (%) | | Suicides | | | | | | | | Specialities |
|-------|---------|-----------|-------------|------|-------|------|-------|------|-------|------|-------|------|-------|--------------|
| | | | | | | Death–n (%) | | Mortality–SMR (95CI) | | Attempts—n | | Thoughts—n | | |
| | | | | Men | Women | Men | Women | Men | Women | Men | Women | Men | Women | |
| Shang 2012 | Taiwan | Asia | 1990–2006 | | | | | | | | | | | No specified |
| Simon 1968 | USA | North America | 1947–1967 | | | | | | | | | | | No specified |
| Stefansson 1991 | Sweden | Europe | 1971–1985 | | | 113 (82) | 25 (19) | 1.82 (1.19, 2.80) | 5.02 (1.67, 15.0) | | | | | No specified |
| Torre 2005 | USA | North America | 1948–1998 | 183 (91) | 18 (11) | 20 (90) | 2 (10) | 1.82 (1.11, 2.82) | 4.95 (0.56, 17.9) | | | | | No specified |
| Ullmann 1991 | USA | North America | 1910–1981 | | | 46 | | 1.48 (0.97, 2.27) | | | | | | No specified |
| Wang 2017 | China | Asia | 2004–2017 | | | 6 (33) | 8 (44) | | | | | | | Dermatologists, emergency, internal medicine, obstetricians, paediatrics, cardiology, neurology, urology, ophthalmology, anaesthesiologists |

## Meta-analysis of the prevalence of deaths by suicide in physicians among all deaths by suicide in the general population

We included nine studies [1,5,15,34,35,36,37,38,82], and we demonstrated a prevalence of 1% (95CI 1, 1) with an important heterogeneity ($I^2$ = 98.0%) (S4 Fig). Insufficient data did not permit meta-regression.

## Meta-analysis of the number of physicians having done suicide attempt among all the physicians

We included five studies [47,57,75,77,85]. The overall effect size was 0.01 (95CI 0.01, 0.02; $p < 0.01$) with an important heterogeneity ($I^2$ = 82.6%) (S5 Fig). Insufficient data did not permit meta-regression.

## Meta-analysis of the number of physicians with suicidal ideation among all the physicians

We included seven studies [74,75,76,77,78,84,85]. The overall effect size was 0.17 (95CI 0.12, 0.21; $p < 0.001$) with an important heterogeneity ($I^2$ = 98.8%) (Fig 11). Insufficient data did not permit meta-regression.

## Other health care workers

As there are few exploitable studies about dental surgeons, nurses and other health-care workers, we didn't treat them in that meta-analysis.

## Discussion

Physicians were an at-risk profession (1.44, 95CI 1.16, 1.72), particularly women-physician (0.67, 95CI 0.19, 1.14; p = 0.007). Some countries had a high risk of suicide (USA vs Rest of the world: 1.34, 95CI 1.28, 1.55; p < 0.001) and rate of suicide in physicians decreased over time, especially in Europe (-0.18, 95CI -0.37, -0.01; p = 0.044). Some specialties were higher risk such as anesthesiologists, psychiatrists, general practitioners and general surgeons. The prevalence of physicians having done suicide attempt among all the physicians were significant (0.01, 95CI 0.01, 0.02; p < 0.001) as the prevalence of physicians with suicidal ideation among all the physicians (0.17, 95CI 0.12, 0.21; p < 0.001). Finally, there were not enough exploitable data about dental surgeons, nurses and other health-care workers which are however some at-risk professions.

### An at-risk profession

The high risk of suicide in physicians might be explained by several putative factors such as psychosocial working environment [18], or specific personality traits of physicians. Psychosocial work environment has been shown in the literature as an important risk factor, doctors being confronted to conflicts with colleagues, lack of cohesive teamwork and social support, leading them individually [88]. Physicians must also routinely face with breaking bad news [89], and are in frequent contact with illness, anxiety, suffering and death. Perfectionism, compulsive attention to detail, exaggerated sense of duty, excessive sense of responsibility, desire to please everyone are appreciates qualities in workplace [90,91] but increased stress and depression [92] and imprison physicians in vicious circle without seek help. They also prevent themselves to ask for help because of the culture of medical education [90,91]. In particular, we demonstrated that women physicians were particularly exposed to suicide, which might be explained by the additional strain imposed on them because of their social roles [11]. In most countries, women still have more at-home responsibilities (education of children, nursing, household care, etc) than men. Combining a full-time job as a physician and those at-home responsibilities might be particularly difficult to manage [11]. Although income gender-inequalities have not been reported in physicians[93,94], some authors suggested that the medical field was mainly dominated by the male gender and reported a poor status integration of women physicians within the profession [7]. It has been shown that female physicians/internships react by imposing themselves an additional pressure to demonstrate their male counterparts that they are as strong, self-sufficient and worthy as them [95].

### Depending on countries

We showed that the risk of suicide was not homogeneous between countries, in line with inequality of job satisfaction among physicians in many countries [96,97]. Indeed, some countries such as Switzerland and Canada reported a high level of job satisfaction for physicians (>75%) [98,99]. In the United States, most obstetrician gynecologists only rated their job satisfaction as moderate [100]. Physician job satisfaction is essential for ensuring the quality and sustainability of health care provision [101,102]. Moreover, career dissatisfaction was associated with burnout and prolonged fatigue among physicians [103]. In most countries, physicians' work conditions underwent frequent mutations, with multiple healthcare reforms initiatives promoting by local governments. Reforms are a necessary compromise between best outcomes on deliveries of care, health economics, and quality of work environment [104, 105].

## With a time effect

There are few data on the evolution of the rate of suicide over time and we were the first to demonstrate that, in some countries such as in Europe the suicide rate among physicians decreased significantly with time but not in the USA. During the past decade, a confluence of forces has changed the practice of medicine in unprecedented ways. Indeed, physicians have seen their autonomy reduced by increased administrative tasks and time pressure [106,107, 108]. In USA, a survey showed that physicians' satisfaction declined over the last 10 years, with less time spent per patient and for private life [13]. US physicians might also be particularly stress [109] because of medical errors that are the third leading cause of death in US [110,111] in a context of economic pressure and relationships with pharmaceutic companies [112,113], religious beliefs [114], access care difficulties for some patients [115], and legal procedure intended against physicians [116] leading them to practice a more defensive medicine [117] misleading patients in overdiagnosis [118]. The World Health Organization global strategy on human resources for health (workforce 2030) promoted the personal and professional rights of health-care workers, including safe and decent working environments [119]. Particularly in Europe, working hours of physicians decreased significantly over the last decades following official instructions such as the European Working Time Directive (EWTD) [14], which may have contributed to a decreased risk of suicides.

## Some specialties are more at-risk

We showed some the most at-risk specialties were anaesthesiologists, psychiatrists, general practitioners and general surgeons. The high risk of suicides in anaesthesiologists [16,41,48, 76] could be explained by an easy access to potentially lethal drugs, a high prevalence of burn-out [120], a high workload with fear of harming patients and organizational burden with poor autonomy, and conflicts with colleagues [121]. For psychiatrists, the high risk of suicides has been linked by stressful and traumatic experiences such as, paradoxically, dealing with suicides of patient [16]. Next to those medical specialties, the general practitioners were an historical at-risk occupation, with moral loneliness, job interfering with family life, constant interruptions both at home and at work, increasing administrative constraints, and high levels of patients' expectations, leading to a low job satisfaction and poor mental health [122,123]. Finally, specialties with life-and-death emergencies, like surgery, are particularly stressful [124, 125,126,127]. For example, it has been shown that intra-operative death increased morbidity in patients operated by the same surgeon in the subsequent 48 hours, with a more pronounced whether the death occurring during emergency surgery [128].

## Suicide attempts and suicidal ideation

Suicide could be regarded as a lengthy process. Little is known about causes and transitions between suicidal ideation / attempted suicide and suicide, as well as about the factors that precipitate or protect against these transitions [129]. Because physicians might be more aware of these characteristics than the general population [75], having suicidal thoughts should be taken particularly seriously in this profession. Suicidal ideation are considered a sensitive and specific indicator of suicide risk [130,131]. Preventive strategies may include improved management of psychiatric disorders, the recognition and treatment of depression and substances abuse [65], but also measures to reduce occupational stress, and restriction of access to means of suicide when doctors are depressed [4,132]. Medical school curriculum should also include programs to increase students' self-confidence, to express their emotional needs, and to teach that anyone may be suicidal–regardless of his status [133]. The preventive approach may

consist of screening, assessment, referral and education, and to destigmatize help-seeking at-risk medical students/physicians [134].

## Suicides in other health-care workers

We highlighted the lack of studies providing data on deaths by suicide and on suicidal risks in nurses and in other health-care workers. However, nurses remained at high-risk of suicide with various stressful factors comparable to those previously described for physicians, such as patients cares, team's conflicts, heavy workload, lack of autonomy, and work-family conflicts [135,136]. As for physicians, some occupational settings were described as particularly stressful, such as working in emergency departments [137], with a high prevalence of shift work [138], exposure to aggressive and violent behavior from patients [139] and from situation relating to trauma, alcohol and intoxications [140]. Our study demonstrated the lack of data on other health-care workers such as pharmacists, dental surgeons, midwives, caregivers and hospital maids. We believe that such data are needed.

## Limitations

Our study has however some limitations. Meta-analyses inherit the limitations of the individual studies of which they are composed: varying quality of studies and multiple variations in study protocols and evaluation. We highlighted that general practitioners were prone to suicide. However, comparisons between specialties may suffer from a major bias such as different number of physicians within each specialty (not the same denominator in statistical analyses—there are more suicides among general practitioners because there are more general practitioners than other individual specialties). All included studies on death by suicide in physicians were retrospective and based on health registers, and thus few studies reported details on occupation such as seniority or characteristics of practice, precluding further analyses necessary for effective preventive strategies. The studies on suicide attempts and suicidal ideation that were based on self-report questionnaire [73,74,75,77] may lack of standardized interviews or specifics criteria for diagnoses psychiatric disorders [125,[141]. Most cross-sectional studies included in our meta-analyses described a bias of self-report such as skipping questions and incomplete information, nondisclosure, and uncertainty regarding timing of questionnaire. Percentage of respondents within those studies may seem low, from 45% [74] to 76% [77], however the response rate was higher than usual [142,143,144,145,146]. The language used in countries with two official languages may also have influenced responses [74]. Only one study questioned physicians on their antidepressant treatment [121], and only one study questioned about a psychiatric disorder [74]. More data is needed regarding physician's health. Finally, none of the studies included specified whether some physicians were retired or not.

## Conclusion

Preventive strategies on the risk of suicides in physicians are strongly needed. Physicians are an at-risk profession of suicide, with a global SMR of 1.44 (95CI 1.16, 1.72), and an important heterogeneity between studies. Women were particularly at risk compared to male physicians. In addition, some countries were with a higher risk of suicide such as USA. Interestingly, the rate of suicide in physicians decreased over time, especially in Europe, suggesting improvements of working conditions of physicians. Some specialties might be at higher risk such as anesthesiologists, psychiatrists, general practitioners and general surgeons. The high prevalence of physicians who committed suicide attempts as well as those with suicidal ideation should benefits for preventive strategies at the workplace. Public health policies must aim at improving social work environment and contribute to screening, assessment, referral, and

destigmatization of suicides in physicians. Finally, the lack of data on other health-care workers suggest implementing studies investigating those occupations who might also be at risk of suicide.

## Supporting information

**S1 Appendix. Details on study characteristics, quality of articles (Figs 2 and 3), method of sampling for markers analysis, inclusion and exclusion criteria, characteristics of participants, outcomes and aims of the studies, and study designs of included articles.**
(DOCX)

**S2 Appendix. PRISMA checklist.**
(DOCX)

**S1 Fig. Meta-regression of percentages of suicide in physicians by group of specialties.**
(TIF)

**S2 Fig. Meta-regression of percentages of suicide in physicians by category of surgical specialties.**
(TIF)

**S3 Fig. Meta-regression of percentages of suicide in physicians by category of medical specialties.**
(TIF)

**S4 Fig. Meta-analysis of prevalence of physicians died by suicide among all the deaths by suicide in the general population.**
(TIF)

**S5 Fig. Meta-analysis of prevalence of physicians having done suicide attempt among all the physicians.**
(TIF)

## Acknowledgments

We wish to thank Richard May for providing assistance in improving the manuscript.

## Author Contributions

**Conceptualization:** Frédéric Dutheil, Claire Aubert, Valentin Navel.

**Data curation:** Frédéric Dutheil, Claire Aubert, Bruno Pereira, Michael Dambrun, Valentin Navel.

**Formal analysis:** Frédéric Dutheil, Claire Aubert, Michael Dambrun, Martial Mermillod.

**Investigation:** Claire Aubert, Valentin Navel.

**Methodology:** Frédéric Dutheil, Claire Aubert, Bruno Pereira, Michael Dambrun, Martial Mermillod, Julien S. Baker, Valentin Navel.

**Project administration:** Frédéric Dutheil, Claire Aubert, Fares Moustafa.

**Resources:** Claire Aubert, Michael Dambrun, Valentin Navel.

**Software:** Frédéric Dutheil, Bruno Pereira.

**Supervision:** Frédéric Dutheil, François-Xavier Lesage.

**Validation:** Frédéric Dutheil, Bruno Pereira, Michael Dambrun, Fares Moustafa, Martial Mermillod, Julien S. Baker, Marion Trousselard, François-Xavier Lesage.

**Visualization:** Frédéric Dutheil, Claire Aubert, Julien S. Baker, Marion Trousselard.

**Writing – original draft:** Frédéric Dutheil, Claire Aubert, Valentin Navel.

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
