## [Decision Letter · Decision Letter 0]

23 Aug 2019

PONE-D-19-21600

Suicide among physicians and health-care workers A systematic review and meta-analysis

PLOS ONE

Dear Dr. Navel,

Thank you for submitting your manuscript to PLOS ONE. After careful consideration, we feel that it has merit but does not fully meet PLOS ONE’s publication criteria as it currently stands. Therefore, we invite you to submit a revised version of the manuscript that addresses the points raised during the review process.

Please address all concerns raised by our reviewer. English editing should be done as pointed out. Moreover, further detailed information and clearer descriptions, especially on methodology, must be required in order to convince our readers on why the authors approach was relevant and valid scientifically.  

We would appreciate receiving your revised manuscript by Oct 07 2019 11:59PM. To enhance the reproducibility of your results, we recommend that if applicable you deposit your laboratory protocols in protocols.io, where a protocol can be assigned its own identifier (DOI) such that it can be cited independently in the future. For instructions see: http://journals.plos.org/plosone/s/submission-guidelines#loc-laboratory-protocols

We look forward to receiving your revised manuscript.

Kind regards,

Takeru Abe, Ph.D

Academic Editor

PLOS ONE

Journal Requirements:

Additional Editor Comments (if provided):

Reviewers' comments:

Reviewer's Responses to Questions

**Comments to the Author**

1. Is the manuscript technically sound, and do the data support the conclusions?

Reviewer #1: Yes

2. Has the statistical analysis been performed appropriately and rigorously? 

Reviewer #1: Yes

3. Have the authors made all data underlying the findings in their manuscript fully available?

Reviewer #1: Yes

4. Is the manuscript presented in an intelligible fashion and written in standard English?

Reviewer #1: No

5. Review Comments to the Author

Reviewer #1: Thank you for the opportunity to review the manuscript, “Suicide among physicians and health-care workers: A systematic review and meta-analysis.” This manuscript conducts an updated meta-analytic review of suicide risk among healthcare works. The authors tackle a topic of need, as demonstrated by the high rates of suicide among healthcare workers, and physicians more specifically. While I believe this manuscript may be of value to the broader literature, there are several spots where more information or discussion would greatly enhance the potential impact. These points, in addition to a few more minor points, are outlined below.

- I appreciate the author’s erring toward brevity in setting up the rational for the current study. However, given the number of, and content of, the study hypotheses, there is a need for more background information. It is not clear why the authors are hypothesizing many of the points that they are. For example, why do they think that females would have greater risk of death by suicide? The general literature demonstrates that men are more likely to die by suicide. Similarly, why do they think that rates by profession will improve overtime?

- Additional information on the inclusion /exclusion criteria and how this impacted study selection is needed. Did studies need to be peer-reviewed, present empirical data, etc. ? Why were studies on medical students excluded but those including interns included?

- Further, it is unclearly why studies needed to include information on both healthcare workers and the general population for inclusion criteria (pg. 5, sentence 4) when not all analyses required this. It would also be helpful to include how many studies were not included per exclusion criteria listed in Figure 1.

- The authors detail 4 different meta-analyses in the statistical considerations questions, but present information on 8 different models. Greater detail of the models in the statistical considerations section would be useful. For example, further explanation of how some of the models are different would be important to include (i.e., meta-analysis of percentage of suicide in physicians by group of specialties vs. that by category of medical specialty). This might be a point of discussion to further delineate in the introduction.

- How many studies were conducted in the US? Currently the figures just note North America, but analyses also target the US.

- Given the significance of gender analyses, it would be useful to have N’s by gender for studies versus just percentage (since we don’t know the overall N for studies).

- Greater information on how the different time periods were handled in analyses would be useful (i.e., some time periods included 30+ years where others were shorter, some were partially overlapping, etc.).

- In the discussion it would be useful if authors discussed in more depth the gender finding. This seems to be a major finding of the paper but it only receives 1-2 lines in the discussion.

- Similarly, the discussion could be enhanced overall by increasing the depth of discussion of findings. For example, when discussing findings related to the US authors discuss career dissatisfaction but don’t really discuss why this might be different in the US.

- Greater discussion of the implications of these findings in the Conclusion section is needed. What do these findings mean? How should be people use this information?

- Generally the Figures were very hard to read, sometimes did not fit on the page, and did not include enough information to stand alone (e..g, Figure 3).

- I would suggest that the authors have someone native to English copy-edit the manuscript, simply to examine verb disagreements, etc.

6. PLOS authors have the option to publish the peer review history of their article (what does this mean?). If published, this will include your full peer review and any attached files.

Reviewer #1: No

---

## [Author Response · Author response to Decision Letter 0]

2 Oct 2019

Dear Editor,

My coauthors and I welcomed the review of our Manuscript PONE-D-19-21600 entitled “Suicide among physicians and health-care workers A systematic review and meta-analysis”. We have addressed the comments of the reviewers in a revised manuscript and enclose a point-by-point response.

Editor Comments

None

[REPLY] Thank you for letting us know that all questions were already included into reviewers’ comments.

Journal Requirements:

[REPLY] Thank you for your comment. The manuscript now follows Journal Requirements.

Reviewers' Comments

Thank you for the opportunity to review the manuscript, “Suicide among physicians and health-care workers: A systematic review and meta-analysis.” This manuscript conducts an updated meta-analytic review of suicide risk among healthcare works. The authors tackle a topic of need, as demonstrated by the high rates of suicide among healthcare workers, and physicians more specifically. While I believe this manuscript may be of value to the broader literature, there are several spots where more information or discussion would greatly enhance the potential impact. These points, in addition to a few more minor points, are outlined below.

[REPLY] Thank you for you positive comment. We have addressed a point-by-point response below.

- I appreciate the author’s erring toward brevity in setting up the rational for the current study. [REPLY] Thank you for you positive comment.

However, given the number of, and content of, the study hypotheses, there is a need for more background information. It is not clear why the authors are hypothesizing many of the points that they are. For example, why do they think that females would have greater risk of death by suicide? The general literature demonstrates that men are more likely to die by suicide. 

[REPLY] Thank you for your relevant comment. The introduction now reads: “Suicide risk was increased in certain occupational groups, especially in medical-related professions [1]. Physicians, and other health-care workers such as nurses [2,3], were considered like high risk group of suicide in different countries [4,5,6], especially for women [6,7,8]. Indeed, despite considerably higher risk of suicides in men than women in the general population [9], female doctors have higher suicide rates than men [10], putatively because of their social family role [95], or a poor status integration within the profession [7].” We also added more details in the discussion.

Similarly, why do they think that rates by profession will improve overtime?

[REPLY] Thank you for your relevant comment. The introduction now reads: “Physicians working conditions varied substantially between countries and over contemporary times, these factors were never investigated in relationships with suicide in physicians. For example, recent years saw tentative to regulate working time of physicians, such as in Europe with its European Working Time Directive (EWTD) [Reference].”

Reference: Temple, J. (2014). Resident duty hours around the globe: where are we now? BMC Medical Education, 14(1), S8. doi:10.1186/1472-6920-14-S1-S8

- Additional information on the inclusion /exclusion criteria and how this impacted study selection is needed. Did studies need to be peer-reviewed, present empirical data, etc. ? 

[REPLY] The methods section now reads: “To be included, articles had to be peer-reviewed and to describe original empirical data on suicides, suicide attempt or suicidal ideation in health-care workers.”

Why were studies on medical students excluded but those including interns included?

[REPLY] The methods section now reads: “Students were excluded because of the difference in responsibilities in comparisons with health-care workers, and because of the existence of previous recent meta-analyses focusing specifically on health-care students [References].”

References:

Puthran, R., Zhang, M. W., Tam, W. W., & Ho, R. C. (2016). Prevalence of depression amongst medical students: a meta-analysis. Medical Education, 50(4), 456-468. doi:10.1111/medu.12962

Rotenstein, L. S., Ramos, M. A., Torre, M., Segal, J. B., Peluso, M. J., Guille, C., . . . Mata, D. A. (2016). Prevalence of Depression, Depressive Symptoms, and Suicidal Ideation Among Medical Students: A Systematic Review and Meta-Analysis. JAMA, 316(21), 2214-2236. doi:10.1001/jama.2016.17324

Witt, K., Boland, A., Lamblin, M., McGorry, P. D., Veness, B., Cipriani, A., . . . Robinson, J. (2019). Effectiveness of universal programmes for the prevention of suicidal ideation, behaviour and mental ill health in medical students: a systematic review and meta-analysis. Evid Based Ment Health, 22(2), 84-90. doi:10.1136/ebmental-2019-300082

Zeng, W., Chen, R., Wang, X., Zhang, Q., & Deng, W. (2019). Prevalence of mental health problems among medical students in China: A meta-analysis. Medicine, 98(18), e15337. doi:10.1097/md.0000000000015337

- Further, it is unclearly why studies needed to include information on both healthcare workers and the general population for inclusion criteria (pg. 5, sentence 4) when not all analyses required this.

[REPLY] Thank you for your relevant comment. We totally agree with you and deleted “in the general population”. The methods section now reads: “To be included, articles had to be peer-reviewed and to describe original empirical data on suicides, suicide attempt or suicidal ideation in health-care workers.

It would also be helpful to include how many studies were not included per exclusion criteria listed in Figure 1.

[REPLY] Thank you for your relevant comment. Figure 1 now includes the number of studies not included per exclusion criteria.

- The authors detail 4 different meta-analyses in the statistical considerations questions, but present information on 8 different models. Greater detail of the models in the statistical considerations section would be useful. For example, further explanation of how some of the models are different would be important to include (i.e., meta-analysis of percentage of suicide in physicians by group of specialties vs. that by category of medical specialty). This might be a point of discussion to further delineate in the introduction.

[REPLY] Thank you for your relevant comment. In fact, we computed four types of meta-analyses: 1) on SMR, 2) on prevalence of suicides among all health-care workers death, 3) on prevalence of suicides among all the death by suicides in the general population, and 4) on suicide attempts and suicidal ideation); but each type might be composed of several meta-analysis in subgroups, such as SMR by sex, SMR by geographic zones, SMR by epochs of time, and SMR by categories of specialties. To facilitate understanding for readers, we have chosen to add the Figures in parenthesis and to give more details. The methods section now reads: “We conducted: 1) meta-analyses on the Standardized Mortality Ratio (SMR) for suicides i.e. the ratio between the observed and expected number of death among physicians, stratified by sex (Fig 4; and Fig 5 for metaregressions), geographic zones (Fig 6), epochs of time, and by categories of specialties (main groups of specialities (Fig 7 and S1 Fig), surgical specialties (Fig 8 and S2 Fig), then medical specialities (Fig 9 and S3 Fig), 2) meta-analyses on the prevalence of health-care workers died by suicide among all health-care workers death (Fig 10), 3) meta-analyses on the prevalence of health-care workers died by suicide among all the deaths by suicide in the general population (S4 Fig), 4) meta-analyses on suicide attempts (S5 Fig) and suicidal ideation (Fig 11).” We added the following sentence in the introduction to emphasize our further objective to compare between specialties: “Some specialties have been suggested to be particularly at risk of suicides [15,16] with occupational factors individualized in different medical or surgical specialties: heavy workload and working hours involved in the job such as long shifts and unpredictable hours (with the sleep deprivation associated) [17], stress of the situations (life and death emergencies) [18], and easy access to a means of committing suicide [19].” 

- How many studies were conducted in the US? Currently the figures just note North America, but analyses also target the US.

[REPLY] Thank you for your relevant comment. We added a new Table 1 that gives details on studies including country.

- Given the significance of gender analyses, it would be useful to have N’s by gender for studies versus just percentage (since we don’t know the overall N for studies).

[REPLY] Thank you for your relevant comment. We added a new Table 1 that gives details on N’s by gender for studies versus just percentage.

- Greater information on how the different time periods were handled in analyses would be useful (i.e., some time periods included 30+ years where others were shorter, some were partially overlapping, etc.).

[REPLY] Thank you for your relevant comment. The statistics section now reads: “When possible (sufficient sample size), meta-regressions were proposed to study relation between prevalence and epidemiological relevant parameters determined according to the literature: sex, geographic zone, epoch of time (for studies with a follow-up over several consecutive years, we based our statistics on the mean year of epoch of time).”

- In the discussion it would be useful if authors discussed in more depth the gender finding. This seems to be a major finding of the paper but it only receives 1-2 lines in the discussion.

[REPLY] Thank you for your relevant comment. We added the following sentences in the discussion: “In particular, we demonstrated that women physicians were particularly exposed to suicide, which might be explained by the additional strain imposed on them because of their social roles [95]. In most countries, women still have more at-home responsibilities (education of children, nursing, household care, etc) than men. Combining a full-time job as a physician and those at-home responsibilities might be particularly difficult to manage [95]. Although income gender-inequalities have not been reported in physicians [97,98], some authors suggested that the medical field was mainly dominated by the male gender and reported a poor status integration of women physicians within the profession [7]. It has been shown that female physicians/internships react by imposing themselves an additional pressure to demonstrate their male counterparts that they are as strong, self-sufficient and worthy as them [99].”.

References:

97. Smith SJ (1990) Income, Housing Wealth and Gender Inequality. Urban Studies 27: 67-88.

98. Finch N (2014) Why are women more likely than men to extend paid work? The impact of work-family life history. Eur J Ageing 11: 31-39.

99. Pospos S, Tal I, Iglewicz A, Newton IG, Tai-Seale M, Downs N, et al. (2019) Gender differences among medical students, house staff, and faculty physicians at high risk for suicide: A HEAR report. Depress Anxiety.

- Similarly, the discussion could be enhanced overall by increasing the depth of discussion of findings. For example, when discussing findings related to the US authors discuss career dissatisfaction but don’t really discuss why this might be different in the US.

[REPLY] Thank you for your relevant comment. The discussion now reads: “There are few data on the evolution of the rate of suicide over time and we were the first to demonstrate that, in some countries such as in Europe the suicide rate among physicians decreased significantly with time but not in the USA. During the past decade, a confluence of forces has changed the practice of medicine in unprecedented ways. Indeed, physicians have seen their autonomy reduced by increased administrative tasks and time pressure [110,111,112]. In USA, a survey showed that physicians’ satisfaction declined over the last 10 years, with less time spent per patient and for private life [13]. US physicians might also be particularly stress [113] because of medical errors that are the third leading cause of death in US [114,115] in a context of economic pressure and relationships with pharmaceutic companies [116,117], religious beliefs [118], access care difficulties for some patients [119], and legal procedure intended against physicians [120] leading them to practice a more defensive medicine [121] misleading patients in overdiagnosis [122]. The World Health Organization global strategy on human resources for health (workforce 2030) promoted the personal and professional rights of health-care workers, including safe and decent working environments [123]. Particularly in Europe, working hours of physicians decreased significantly over the last decades following official instructions such as the European Working Time Directive (EWTD) [14], which may have contributed to a decreased risk of suicides.”

References:

113. Leape LL (1994) Error in medicine. Jama 272: 1851-1857.

114. Makary MA, Daniel M (2016) Medical error-the third leading cause of death in the US. Bmj 353: i2139.

115. Anderson JG, Abrahamson K (2017) Your Health Care May Kill You: Medical Errors. Stud Health Technol Inform 234: 13-17.

116. Mitchell AP, Winn AN, Lund JL, Dusetzina SB (2019) Evaluating the Strength of the Association Between Industry Payments and Prescribing Practices in Oncology. Oncologist 24: 632-639.

117. Wazana A (2000) Physicians and the pharmaceutical industry: is a gift ever just a gift? Jama 283: 373-380.

118. Korup AK, Sondergaard J, Lucchetti G, Ramakrishnan P, Baumann K, Lee E, et al. (2019) Religious values of physicians affect their clinical practice: A meta-analysis of individual participant data from 7 countries. Medicine (Baltimore) 98: e17265.

119. Dickman SL, Himmelstein DU, Woolhandler S (2017) Inequality and the health-care system in the USA. Lancet 389: 1431-1441.

120. Berlin L (2017) Medical errors, malpractice, and defensive medicine: an ill-fated triad. Diagnosis (Berl) 4: 133-139.

121. Studdert DM, Mello MM, Sage WM, DesRoches CM, Peugh J, Zapert K, et al. (2005) Defensive medicine among high-risk specialist physicians in a volatile malpractice environment. Jama 293: 2609-2617.

122. Chiolero A, Paccaud F, Aujesky D, Santschi V, Rodondi N (2015) How to prevent overdiagnosis. Swiss Med Wkly 145: w14060.

- Greater discussion of the implications of these findings in the Conclusion section is needed. What do these findings mean? How should be people use this information?

[REPLY] Thank you for your relevant comment. We added implications of these findings in the Conclusion. The conclusion now reads: “Preventive strategies on the risk of suicides in physicians are strongly needed. Physicians are an at-risk profession of suicide, with a global SMR of 1.44 (95CI 1.16, 1.72), and an important heterogeneity between studies. Women were particularly at risk compared to male physicians. In addition, some countries were with a higher risk of suicide such as USA. Interestingly, the rate of suicide in physicians decreased over time, especially in Europe, suggesting improvements of working conditions of physicians. Some specialties might be at higher risk such as anesthesiologists, psychiatrists, general practitioners and general surgeons. The high prevalence of physicians who committed suicide attempts as well as those with suicidal ideation should benefits for preventive strategies at the workplace. Public health policies must aim at improving social work environment and contribute to screening, assessment, referral, and destigmatization of suicides in physicians. Finally, the lack of data on other health-care workers suggest implementing studies investigating those occupations who might also be at risk of suicide.”.

- Generally the Figures were very hard to read, sometimes did not fit on the page, and did not include enough information to stand alone (e.g., Figure 3).

[REPLY] Thank you for your relevant comment. We agree that there was a need to provide further details on each included articles. In order to keep Figures as simple as possible, we added a new Table 1 with details (including gender and country) for each study. Figure 3 is common in meta-analysis as a summary of risks of bias (e.g. doi: 10.1016/j.jtos.2019.06.004 impact factor 9.1, doi: 10.1001/jama.2018.20578 impact factor 51), in order to give more confidence on results of our meta-analysis. However, Figure 3 can be proposed as a supplementary material on request.

- I would suggest that the authors have someone native to English copy-edit the manuscript, simply to examine verb disagreements, etc.

[REPLY] We wish to thank Richard May, native English, for providing assistance in improving the manuscript.

We hope our work will be considered favorably and look forward to hearing from you.

Sincerely yours,

---

## [Decision Letter · Decision Letter 1]

4 Nov 2019

PONE-D-19-21600R1

Suicide among physicians and health-care workers: A systematic review and meta-analysis

PLOS ONE

Dear Dr. Navel,

Thank you for submitting your manuscript to PLOS ONE. After careful consideration, we feel that it has merit but does not fully meet PLOS ONE’s publication criteria as it currently stands. Therefore, we invite you to submit a revised version of the manuscript that addresses the points raised during the review process.

There would be a few points to be clarified. Please address all comments by our reviewer.

We would appreciate receiving your revised manuscript by Dec 19 2019 11:59PM. To enhance the reproducibility of your results, we recommend that if applicable you deposit your laboratory protocols in protocols.io, where a protocol can be assigned its own identifier (DOI) such that it can be cited independently in the future. For instructions see: http://journals.plos.org/plosone/s/submission-guidelines#loc-laboratory-protocols

We look forward to receiving your revised manuscript.

Kind regards,

Takeru Abe, Ph.D

Academic Editor

PLOS ONE

Reviewers' comments:

Reviewer's Responses to Questions

**Comments to the Author**

1. If the authors have adequately addressed your comments raised in a previous round of review and you feel that this manuscript is now acceptable for publication, you may indicate that here to bypass the “Comments to the Author” section, enter your conflict of interest statement in the “Confidential to Editor” section, and submit your "Accept" recommendation.

Reviewer #1: (No Response)

2. Is the manuscript technically sound, and do the data support the conclusions?

Reviewer #1: Yes

3. Has the statistical analysis been performed appropriately and rigorously? 

Reviewer #1: Yes

4. Have the authors made all data underlying the findings in their manuscript fully available?

Reviewer #1: Yes

5. Is the manuscript presented in an intelligible fashion and written in standard English?

Reviewer #1: Yes

6. Review Comments to the Author

Reviewer #1: Thank you for the opportunity to review the revised manuscript, “Suicide among physicians and health-care workers: A systematic review and meta-analysis.” I thank the authors for their thoughtful and thorough response to reviewers. I believe the manuscript is much improved. Only two minor points remain.

- The authors added the sentence “For example, recent years saw tentative to regulate working time of physicians, such as in Europe with its European Working Time Directive (EWTD).” I believe there may be a word missing following tentative.

- Thank you for the added explanation regarding the exclusion of students. However, what about the relevance to interns (some fields consider interns as students, while others do not)? Are they expected to have responsibilities that do not reflect traditional students? Were they included in the previous meta-analysis?

- Thank you for clarifying the inclusion criteria regarding information on healthcare workers vs. the general population in the methods section. However, did this influence the studies included? That is, were studies that did not information on the general population excluded from the review? If so, this would suggest it may be necessary to re-review the excluded studies.

7. PLOS authors have the option to publish the peer review history of their article (what does this mean?). If published, this will include your full peer review and any attached files.

Reviewer #1: No

---

## [Author Response · Author response to Decision Letter 1]

13 Nov 2019

Dear Editor,

My coauthors and I welcomed the review of our Manuscript PONE-D-19-21600 entitled “Suicide among physicians and health-care workers A systematic review and meta-analysis”. We have addressed the comments of the reviewers in a revised manuscript and enclose a point-by-point response.

Review Comments to the Author

Thank you for the opportunity to review the revised manuscript, “Suicide among physicians and health-care workers: A systematic review and meta-analysis.” I thank the authors for their thoughtful and thorough response to reviewers. I believe the manuscript is much improved. Only two minor points remain.

[REPLY] Thank you for your positive comment.

- The authors added the sentence “For example, recent years saw tentative to regulate working time of physicians, such as in Europe with its European Working Time Directive (EWTD).” I believe there may be a word missing following tentative.

[REPLY] Thank you for your comment. The sentence now reads: “For example, there were tentative to regulate working time of physicians over the recent years, such as in Europe with its European Working Time Directive (EWTD).”

- Thank you for the added explanation regarding the exclusion of students. However, what about the relevance to interns (some fields consider interns as students, while others do not)? Are they expected to have responsibilities that do not reflect traditional students? Were they included in the previous meta-analysis?

[REPLY] Thank you for your comment. We included internship students in our meta-analysis because previous meta-analyses did not include interns (medical students included were year 1 to 5 or 6 in all meta-analyses on prevalence of suicids or suicidal ideations – Puthran et al. 2016 Rotenstein et al. 2016 and Zeng et al. 2019) and because they could have similar responsibilities to senior practitioners. We added the following sentence within the Methods section: “Students were excluded because of the difference in responsibilities in comparisons with health-care workers, and because of the existence of previous recent meta-analyses focusing specifically on health-care students [21,22,23,24]; we included interns because they were not included in the aforementioned meta-analyses on prevalence of suicides, suicide attempts or suicidal ideation, and because they could have similar responsibilities to senior practitioners.”

- Thank you for clarifying the inclusion criteria regarding information on healthcare workers vs. the general population in the methods section. However, did this influence the studies included? That is, were studies that did not information on the general population excluded from the review? If so, this would suggest it may be necessary to re-review the excluded studies.

[REPLY] Thank you for your comment. It did not influence the studies included as data in the general population were not mandatory (it was needed only for some meta-analysis for between groups comparison: health care workers versus general population). We added the following sentence within the Methods section: “When data were available, we also collected data from a control group (such as general population) for comparisons purposes.”

---

## [Decision Letter · Decision Letter 2]

26 Nov 2019

Suicide among physicians and health-care workers: A systematic review and meta-analysis

PONE-D-19-21600R2

Dear Dr. Navel,

We are pleased to inform you that your manuscript has been judged scientifically suitable for publication and will be formally accepted for publication once it complies with all outstanding technical requirements.

With kind regards,

Takeru Abe, Ph.D

Academic Editor

PLOS ONE

Additional Editor Comments (optional):

Just for clarification to you and our reviewer, I just noted below.

In your response to reviewer's comments, you described:

[REPLY] Thank you for your comment. The sentence now reads: “For example, there

were tentative to regulate working time of physicians over the recent years, such as in

Europe with its European Working Time Directive (EWTD).”

However, the sentence in the manuscript reads below:

For example, there were attempts in recent years to regulate physicians’ working time, e.g. the European Working Time Directive (EWTD) in Europe.

Reviewers' comments:

Reviewer's Responses to Questions

**Comments to the Author**

1. If the authors have adequately addressed your comments raised in a previous round of review and you feel that this manuscript is now acceptable for publication, you may indicate that here to bypass the “Comments to the Author” section, enter your conflict of interest statement in the “Confidential to Editor” section, and submit your "Accept" recommendation.

Reviewer #1: All comments have been addressed

2. Is the manuscript technically sound, and do the data support the conclusions?

Reviewer #1: Yes

3. Has the statistical analysis been performed appropriately and rigorously? 

Reviewer #1: Yes

4. Have the authors made all data underlying the findings in their manuscript fully available?

Reviewer #1: Yes

5. Is the manuscript presented in an intelligible fashion and written in standard English?

Reviewer #1: Yes

6. Review Comments to the Author

Reviewer #1: (No Response)

7. PLOS authors have the option to publish the peer review history of their article (what does this mean?). If published, this will include your full peer review and any attached files.

Reviewer #1: No

---

## [Editor Report · Acceptance letter]

3 Dec 2019

PONE-D-19-21600R2 

Suicide among physicians and health-care workers: A systematic review and meta-analysis 

Dear Dr. Navel:

I am pleased to inform you that your manuscript has been deemed suitable for publication in PLOS ONE. Congratulations! Your manuscript is now with our production department. 

With kind regards,

on behalf of

Dr. Takeru Abe 

Academic Editor

PLOS ONE